# Evaluating the associations and predictive performance of triglyceride-glucose index and related indicators for chronic diseases in a Chinese cohort

Hongli Liu[1]☯, Xinmei Mao[1]☯, Xuechen Wang[1]☯, Dan Xu[2], Ting Wen[1], Jipeng Li[3]*

1 Outpatient Department(OPD), The General Hospital of Western Theater Command, Chengdu, China, 2 Department of Radiology, China MCC5 Group Corp.Ltd.Hospital, Chengdu, China, 3 Department of Ultrasound, The General Hospital of Western Theater Command, Chengdu, China

☯ These authors contributed equally to this work.
* Ljp_712@outlook.com

## Abstract

### Background

Triglyceride and glucose (TyG) indices have been used as predictors of several chronic diseases. However, there is currently a lack of research that can comprehensively reflect the impact of TyG-related indicators on chronic diseases in middle-aged and elderly populations. The aim of this study was to investigate the relationship of TyG and its related indicators with chronic diseases and their time-dependent predictive ability in the elderly.

### Study design

Retrospective observational cohort study using China Health and Retirement Longitudinal Study (CHARLS) 2011–2020 data.

### Methods

Based on longitudinal data obtained from the CHARLS from 2011 to 2020, a total of 12,966 participants were included in the study. Participants were stratified into three groups according to their TyG index. Pearson correlation coefficient and Cox model are used to assess the relationship between the TyG index, its parameters, and common chronic diseases, while Harrell's C-index is used to evaluate their risk prediction capability.

### Results

The TyG index and its related indicators exhibit a positive dose-response relationship with the risk of diabetes, heart disease, dyslipidemia, hypertension, and stroke, while

**Data availability statement:** All relevant data are within the manuscript and its Supporting Information files.

**Funding:** The author(s) received no specific funding for this work.

**Competing interests:** The authors have declared that no competing interests exist.

demonstrating a negative dose-response relationship with digestive system diseases. Harrell's C-index results indicated that TyG-WC demonstrates superior predictive performance overall.

## Conclusion

The TyG index and its related indicators are significantly correlated with newly onset emerging chronic diseases, with TyG-WC exhibits superior risk prediction performance.

## Introduction

Chronic diseases, defined as conditions lasting a year or more that require ongoing medical attention or limit daily activities, include heart disease, cancer, diabetes, depression, and other serious illnesses [1]. As the largest population in the world, China is facing significant challenges due to its rapidly aging population, which has become a pressing social issue [2]. One of the most critical consequences of this demographic shift is the sustained increase in the prevalence of chronic diseases [3]. By 2040, the number of individuals aged 60 and above in China is projected to reach 402 million, accounting for 28% of the total population, with more than 75% of them expected to suffer from at least one chronic disease [4]. Moreover, the risk of developing multiple chronic conditions rises significantly with age [5]. Therefore, effective prevention and management of chronic diseases in the elderly are crucial for alleviating the health, social, psychological, and economic burdens associated with these conditions.

Insulin plays a key role in the pathogenesis of various chronic diseases by influencing blood glucose levels, cell division, and the metabolism of fats and proteins [6]. Under normal physiological conditions, elevated plasma glucose levels can lead to increased insulin secretion and circulating insulin levels, thereby stimulating glucose transfer to peripheral tissues and inhibiting liver glucose production [7]. However, when insulin function is impaired, it results in a pathological state known as "insulin resistance" (IR), characterized by reduced insulin-mediated glucose control, diminished glucose utilization, abnormal lipid accumulation, and heightened lipid degradation in adipocytes [8]. IR is clinically associated with several metabolic and chronic diseases, including obesity, type 2 diabetes and its complications, non-alcoholic fatty liver disease, tumors, and cardiovascular disease [8,9]. Furthermore, cellular senescence, often observed in age-related chronic diseases, is believed to be linked to IR, with both conditions mutually reinforcing each other [10]. Addressing IR is therefore essential for preventing and managing these chronic diseases.

Despite its importance, the relationship between IR and chronic diseases has not been fully elucidated and remains a topic of ongoing debate. Previous studies have utilized the homeostasis model assessment (HOMA) as a relatively simple and reliable method for assessing IR [11]. More recently, the Triglyceride-Glucose (TyG) index, an IR marker derived from fasting triglyceride and blood glucose levels, has

become a simple and reliable alternative to HOMA [12–15]. The TyG index is now widely used to assess the association between IR and chronic diseases, including chronic kidney disease [16], chronic lung disease [17], osteoarthritis [18], and non-alcoholic fatty liver disease [19]. Evidence suggests that a higher TyG index is associated with an increased risk of chronic disease onset [20–25]. In addition to the effects of metabolic biomarkers such as triglycerides and glucose, disorders of lipid metabolism also play an important role in the adverse consequences caused by IR. Recent studies have shown that combining the TyG index with obesity parameters (such as body mass index [BMI], waist circumference [WC], and waist-to-height ratio [WHtR]) can enhance the identification of IR. Modified TyG indices that integrate these two groups of parameters have also been applied to various diseases [26–28]. Surveys on chronic diseases in the elderly also have consistently recognized the TyG index and related parameters as valuable predictive indicators [21,29,30]. Maintaining a healthy index can help prevent or delay the development of chronic diseases, but the specific association needs to be investigated, because too high or too low can lead to an increased risk of disease. Therefore, our study aims to investigate the association between TyG and its related parameters and chronic diseases in middle-aged and elderly people in China using data from the China Health and Retirement Longitudinal Study (CHARLS) database from 2011 to 2020.

## Data and methods

### Data source and preprocessing

The data of this study were derived from the CHARLS, a national longitudinal survey targeting Chinese middle-aged and elderly populations aged 45 years and above, which has been approved by the Ethics Review Committee of Peking University (IRB00001052–11015). The study protocol was approved by the institutional ethics committee. Informed consent was waived due to the use of de-identified public databases. CHARLS employed a multistage probability proportional to size (PPS) sampling method: in the first stage, 150 counties/districts were selected from 28 provinces (excluding Tibet) through stratification by region, urban-rural division, and per capita GDP; in the second stage, three villages/neighborhood committees were randomly selected from each sampling unit, ultimately covering 450 primary sampling units. Using the CHARLS Geographic Information System (Charls-GIS), a household roster was constructed, with one member aged ≥45 and their spouse (if applicable) randomly selected from each household as respondents. After adjustment for sampling weights, the sample characteristics were consistent with China's national census data, demonstrating national representativenes [31].

This study utilized the latest CHARLS data collected in September 2020 and released in November 2023 (containing five waves of longitudinal data from 2011 to 2020). The data lag primarily resulted from quality control procedures for large-scale biometric measurements. We included participants who completed all five survey waves and excluded those lost to follow-up or meeting any of the following criteria: missing TyG or TyG-BMI indices; > 5% missing values in key variables. Multiple imputation was performed using the mice package in R, where a predictive matrix was created with imputation methods specified according to variable types. Outliers in continuous variables were removed using a function based on normal distribution and the 3σ criterion (S1 Fig).

### Baseline characteristics and correlation analysis

Participants were divided into three groups based on TyG values: Group 1 (TyG < 8.36), Group 2 (8.36 ≤ TyG < 8.87), and Group 3 (TyG ≥ 8.87) [27,32]. The TyG index was calculated as Ln (triacylglycerol (mg/dL) × fasting blood glucose (mg/dL)/2). Baseline characteristics were summarized using the gtsummary package in R, presenting categorical variables as counts and percentages, and continuous variables as means and standard deviations. We calculated obesity-related scores including WC, WHtR, TyG-WC, and TyG-WHtR [33]. Correlation analysis between various diseases and these scores was performed using the phi coefficient for categorical variables and Pearson correlation for continuous variables. Results were visualized using the corrplot package.

## Dose-response relationship analysis

We used data from the CHARLS to calculate the onset time for each disease in the disease list (hibpe, diabe, hearte, stroke, dyslipe, and digeste). For each individual, we determined if the disease was present at the baseline survey. If not, we found the first survey where the disease was reported and calculated the time difference in years and months from the baseline survey. If the disease never occurred, we used the last survey's year and month. The calculated disease onset times were then added to the processed data frame.

To investigate the dose-response relationships between various predictors (TyG, BMI, TyG-BMI, WC, TyG-WC, WHtR, and TyG-WHtR) and diseases, we used restricted cubic spline (RCS) analysis. For each predictor-disease combination, we fitted a Cox regression model adjusting for relevant covariates (age, gender, rural, edu, marry, smokev, drinkev, and conditionally hibpe and diabe). We then fitted an RCS model with 3 knots using the rms package's cph() function. We created prediction plots using the Predict function, setting the reference point to the median of the predictor variable. The predicted hazard ratios and their 95% confidence intervals were plotted using ggplot2, with the median as the reference line.

## Cox proportional hazards model analysis

Univariate Cox proportional hazards regression analysis was conducted to evaluate the impact of predictive factors on chronic disease risk. The TyG index was divided into tertiles. Four models with increasing levels of adjustment were used: Model 1 adjusted for age and gender; Model 2 additionally adjusted for marital status, residential area, education level, smoking status, and alcohol consumption. To address potential residual confounding, we conducted several sensitivity analyses with additional adjustments when data were available. Hazard ratio, 95% confidence interval (CI), and p-value were estimated for each predictor-disease combination.

Since the fundamental assumption of the Cox model is the proportional hazards assumption—that is, the influence of each predictor on the outcome variable remains relatively constant throughout the observation period—we tested this assumption for all models. Specifically, for each combination of predictor variables (TyG, TyG-BMI, WC, TyG-WC, WHtR, TyG-WHR, and BMI) and six chronic diseases (hypertension, diabetes, cardiovascular disease, stroke, dyslipidemia, and digestive system diseases), we used the standard test based on Schoenfeld residuals for evaluation. This method assesses whether the effect of a predictor changes over time by testing the correlation between residuals and time; a p-value greater than 0.05 indicates that the proportional hazards assumption is satisfied.

For predictor-disease combinations that violated the proportional hazards assumption, we incorporated interaction terms between predictors and time (time-dependent covariates) into the models to adjust for time-varying effects. This approach allows hazard ratios of predictors to change over time, enabling more accurate risk estimation. For example, in the model of the relationship between TyG-WC and diabetes, which significantly violated the proportional hazards assumption ($P < 0.001$), we added an interaction term between TyG-WC and time to reflect its time-varying effect.

## Assessment of model calibration and discriminative ability

In terms of risk prediction capability assessment, we used Harrell's C-index, which measures a model's ability to distinguish between high-risk and low-risk individuals. The index ranges from 0.5 (no predictive ability) to 1.0 (perfect prediction).

To comprehensively evaluate the performance of predictive models, we adopted a two-pronged assessment strategy: discrimination and calibration. Discrimination assessment was primarily measured by Harrell's C-index and its 95% CI, which evaluates the model's ability to distinguish between individuals with and without the event. The C-index ranges from 0.5 to 1.0, where 0.5 indicates no predictive ability (equivalent to random guessing), and 1.0 indicates perfect prediction. We defined a C-index $\geq 0.8$ as "excellent discrimination", $0.7 \leq C < 0.8$ as "good discrimination", $0.6 \leq C < 0.7$ as "acceptable discrimination", and $C < 0.6$ as "poor discrimination".

Model calibration was evaluated using two methods: (1) Calibration slope and intercept, derived by dividing predicted risks into deciles and calculating the linear regression between the average predicted probability and the observed event rate in each risk group. An ideal calibration slope should be close to 1.0, and the intercept close to 0, indicating high consistency between predicted probabilities and observed event rates. (2) Hosmer-Lemeshow test, used to assess the goodness-of-fit between predicted probabilities and observed events, where $p > 0.05$ indicates good model calibration. All calibration analyses were performed at the reference follow-up time point (the median follow-up time for each disease).

## Results

### Characteristics of the study population

In this study, a total of 12,966 participants were recruited and divided into three groups based on the distribution of the TyG index, ranging from low to high, the average values were 8.05, 8.61, and 9.28 respectively (Table 1). From the perspective of social demographic characteristics, the average ages of Group 1 to Group 3 are 64.60, 65.00, and 64.81, respectively. The majority of them are urban residents, but the proportion gradually decreases, while the proportion of males gradually increases. In addition, the educational level of the study population is generally low, as more than 60% of the participants have no formal education or only completed primary or secondary education. In terms of marital status, more than 80% of the participants are married. From the perspective of health indicators and biochemical parameters, as the TyG index increases, the proportion of males rises, along with intergroup weight, WC, BMI, and the TyG BMI index. Most participants had no history of diabetes or other chronic diseases in Group 1; however, in Groups 2 and 3 with higher TyG index, the prevalence of hypertension, diabetes, dyslipidemia, heart disease, stroke, arthritis, and memory disorder showed an increasing trend with rising TyG levels. In terms of biochemical parameters, the group with a higher TyG index exhibited elevated levels of white-blood-cell, blood-platelet-count, glucose, total-cholesterol, triglycerides, c-reaction-protein, glycosylated-hemoglobin, uric-acid, hematocrit-value, hemoglobin, systolic-pressure and diastolic-pressure, while mean-corpuscular-volume, blood-urea-nitrogen, high-density-lipoprotein-cholesterol and total-metabolic-output levels decreased as TyG levels increased.

Due to China's large population base and significant geographical differences, which may also influence chronic disease status, we further analyzed the differences in demographic and clinical characteristics between urban and rural residents (Table 2), as well as the distribution of chronic disease prevalence rates among participants from various provinces in China, including hypertension, diabetes, and cardiovascular diseases (Table 3). The results showed significant differences ($p < 0.05$) between urban and rural populations in sex, education level, smoking history, BMI, diabetes, pulmonary diseases, dyslipidemia, cardiovascular diseases, and osteoarthritis (Table 2). Analyses of different provinces revealed significant regional variations in the distribution of chronic diseases across China. For example, the average prevalence rates were relatively high in Beijing (38.2%), Inner Mongolia (38.5%), Tianjin (36.6%), and Hebei (35.3%), while those in Guangdong (14.7%) and Fujian (16.1%) were relatively low (Table 3). Additionally, two geographical distribution maps were created: a map showing the distribution of participant numbers across provinces (Fig 1) and a geographic distribution map of average chronic disease prevalence rates (Fig 2). These visualizations intuitively demonstrate the national distribution characteristics of the CHARLS dataset and the geographic patterns of diseases.

We further investigated the relationship between chronic diseases and the TyG index along with its related indicators (Fig 3). The TyG index showed weak correlations with individual obesity indicators (WC, BMI, WHtR; $r = 0.34$–$0.36$). However, when combined with these obesity measures to form modified indices (TyG-BMI, TyG-WC, and TyG-WHtR), the correlations significantly strengthened ($r = 0.63$–$0.69$), reflecting an intrinsic relationship between TyG and the composite indices. Although these indices demonstrated statistically significant associations with multiple chronic diseases ($p < 0.05$), the strength of correlation was generally modest. Taking hypertension as an example, its correlation coefficients with TyG-WC and TyG-WHtR were relatively higher ($r = 0.27$), yet still fell within the weak correlation range – despite showing

**Table 1. Characteristics of 12966 participants according to baseline TyG levels.**

| Characteristic | | Group 1 (n=4,428) | Group 2 (n=4,379) | Group 3 (n=4,159) |
|---|---|---|---|---|
| Age, years, mean (SD) | | 64.60 (10.34) | 65.00 (9.89) | 64.81 (9.46) |
| Gender, n (%) | Male | 2,140 (48.3%) | 2,496 (57.0%) | 2,454 (59.0%) |
| | Female | 2,288 (51.7%) | 1,883 (43.0%) | 1,705 (41.0%) |
| Residence, n (%) | Rural | 1,423 (32.1%) | 1,677 (38.3%) | 1,855 (44.6%) |
| | Urban | 3,005 (67.9%) | 2,702 (61.7%) | 2,304 (55.4%) |
| Education, n (%) | Uneducated | 2,067 (46.7%) | 2,041 (46.6%) | 1,844 (44.3%) |
| | Primary | 1,041 (23.5%) | 983 (22.4%) | 927 (22.3%) |
| | Secondary | 889 (20.1%) | 873 (19.9%) | 901 (21.7%) |
| | Third | 431 (9.73%) | 482 (11.0%) | 487 (11.7%) |
| Marry, n (%) | No | 822 (18.6%) | 787 (18.0%) | 739 (17.8%) |
| | Yes | 3,606 (81.4%) | 3,592 (82.0%) | 3,420 (82.2%) |
| Smoking, n (%) | No | 2,401 (54.2%) | 2,630 (60.1%) | 2,535 (61.0%) |
| | Yes | 2,027 (45.8%) | 1,749 (39.9%) | 1,624 (39.0%) |
| Drinking, n (%) | No | 2,270 (51.3%) | 2,480 (56.6%) | 2,334 (56.1%) |
| | Yes | 2,158 (48.7%) | 1,899 (43.4%) | 1,825 (43.9%) |
| Height, m, mean (SD) | | 1.58 (0.08) | 1.58 (0.08) | 1.58 (0.09) |
| Weight, kg, mean (SD) | | 56.42 (9.90) | 59.23 (10.73) | 63.09 (11.10) |
| Waist, cm, mean (SD) | | 82.03 (9.20) | 85.79 (9.92) | 90.44 (9.50) |
| BMI, kg/m², mean (SD) | | 22.43 (3.22) | 23.72 (3.47) | 25.20 (3.51) |
| Hypertension, n (%) | No | 3,003 (67.8%) | 2,603 (59.4%) | 2,026 (48.7%) |
| | Yes | 1,425 (32.2%) | 1,776 (40.6%) | 2,133 (51.3%) |
| Diabetes, n (%) | No | 4,166 (94.1%) | 3,894 (88.9%) | 3,268 (78.6%) |
| | Yes | 262 (5.92%) | 485 (11.1%) | 891 (21.4%) |
| Cancer, n (%) | No | 4,311 (97.4%) | 4,244 (96.9%) | 4,036 (97.0%) |
| | Yes | 117 (2.64%) | 135 (3.08%) | 123 (2.96%) |
| Lunge, n (%) | No | 3,721 (84.0%) | 3,623 (82.7%) | 3,523 (84.7%) |
| | Yes | 707 (16.0%) | 756 (17.3%) | 636 (15.3%) |
| Dyslipidemia, n (%) | No | 3,706 (83.7%) | 3,325 (75.9%) | 2,620 (63.0%) |
| | Yes | 722 (16.3%) | 1,054 (24.1%) | 1,539 (37.0%) |
| Heart disease, n (%) | No | 3,597 (81.2%) | 3,328 (76.0%) | 3,049 (73.3%) |
| | Yes | 831 (18.8%) | 1,051 (24.0%) | 1,110 (26.7%) |
| Stroke, n (%) | No | 4,147 (93.7%) | 4,063 (92.8%) | 3,745 (90.0%) |
| | Yes | 281 (6.35%) | 316 (7.22%) | 414 (9.95%) |
| Psyche, n (%) | No | 4,296 (97.0%) | 4,250 (97.1%) | 4,017 (96.6%) |
| | Yes | 132 (2.98%) | 129 (2.95%) | 142 (3.41%) |
| Arthritis, n (%) | No | 2,730 (61.7%) | 2,595 (59.3%) | 2,414 (58.0%) |
| | Yes | 1,698 (38.3%) | 1,784 (40.7%) | 1,745 (42.0%) |
| Liver disease, n (%) | No | 4,109 (92.8%) | 4,065 (92.8%) | 3,863 (92.9%) |
| | Yes | 319 (7.20%) | 314 (7.17%) | 296 (7.12%) |
| Kidney disease, n (%) | No | 3,965 (89.5%) | 3,928 (89.7%) | 3,695 (88.8%) |
| | Yes | 463 (10.5%) | 451 (10.3%) | 464 (11.2%) |
| Digestive disease, n (%) | No | 2,987 (67.5%) | 2,940 (67.1%) | 2,843 (68.4%) |
| | Yes | 1,441 (32.5%) | 1,439 (32.9%) | 1,316 (31.6%) |
| Asthma, n (%) | No | 4,124 (93.1%) | 4,036 (92.2%) | 3,862 (92.9%) |
| | Yes | 304 (6.87%) | 343 (7.83%) | 297 (7.14%) |

*(Continued)*

**Table 1.** (Continued)

| Characteristic | | Group 1 (n = 4,428) | Group 2 (n = 4,379) | Group 3 (n = 4,159) |
|---|---|---|---|---|
| Memory disorder, n (%) | No | 4,212 (95.1%) | 4,140 (94.5%) | 3,897 (93.7%) |
| | Yes | 216 (4.88%) | 239 (5.46%) | 262 (6.30%) |
| WBC,/L, mean (SD) | | 5.63 (1.56) | 5.93 (1.59) | 6.21 (1.61) |
| MCV, fL, mean (SD) | | 91.92 (6.96) | 91.79 (6.60) | 91.50 (6.36) |
| platelet,/L, mean (SD) | | 199.55 (62.63) | 204.89 (63.60) | 208.71 (65.78) |
| BUN, mmol/L, mean (SD) | | 15.61 (4.28) | 15.22 (4.18) | 14.96 (4.00) |
| Glucose, mg/dL, mean (SD) | | 90.32 (12.22) | 98.07 (15.01) | 111.06 (23.61) |
| Creatinine, mg/dL, mean (SD) | | 0.78 (0.18) | 0.78 (0.17) | 0.78 (0.18) |
| Cholesterol, mg/dL, mean (SD) | | 171.74 (30.39) | 185.01 (32.32) | 194.02 (34.07) |
| Triglyceride, mg/dL, mean (SD) | | 72.16 (15.79) | 114.64 (21.63) | 206.80 (65.90) |
| HDL-C, mg/dL, mean (SD) | | 55.82 (11.29) | 51.34 (10.64) | 46.36 (9.49) |
| LDL-C, mg/dL, mean (SD) | | 98.73 (26.25) | 109.05 (28.12) | 106.49 (30.09) |
| CRP, mg/dL, mean (SD) | | 1.66 (2.48) | 1.88 (2.47) | 2.49 (2.51) |
| HbA1c, %, mean (SD) | | 5.57 (0.50) | 5.68 (0.57) | 5.90 (0.73) |
| Uric acid, mg/dL, mean (SD) | | 4.50 (1.20) | 4.74 (1.27) | 5.11 (1.30) |
| Hematocrit, %, mean (SD) | | 40.92 (5.01) | 41.46 (4.90) | 41.83 (4.89) |
| Hemoglobin, g/dL, mean (SD) | | 13.61 (1.73) | 13.79 (1.70) | 13.93 (1.70) |
| Cystatin C, mg/L, mean (SD) | | 0.84 (0.19) | 0.87 (0.20) | 0.87 (0.19) |
| Totmet, MET, mean (SD) | | 5,931.26 (5,740.31) | 5,438.53 (5,515.70) | 5,014.03 (5,223.15) |
| TyG, mean (SD) | | 8.05 (0.23) | 8.61 (0.15) | 9.28 (0.32) |
| TyG_BMI, mean (SD) | | 180.58 (27.16) | 203.70 (30.46) | 233.83 (35.03) |
| SBP, mmHg, mean (SD) | | 123.97 (19.42) | 126.58 (18.98) | 130.30 (18.82) |
| DBP, mmHg, mean (SD) | | 73.00 (11.11) | 74.67 (10.95) | 76.72 (11.03) |
| Cesd10, scores, mean (SD) | | 8.90 (6.38) | 8.98 (6.47) | 8.68 (6.32) |

SD, standard deviation; WBC, white blood cell; MCV, mean corpuscular volume; BUN, blood urea nitrogen; HDL-C, high density lipoprotein cholesterol; LDL-C, low density lipoprotein cholesterol; CRP, C reaction protein; HbA1c, baseline hemoglobin A1c; Totmet, total metabolic equivalent of task; TyG, triglyceride glucose index; BMI, body mass index; SBP, systolic blood pressure; DBP, diastolic blood pressure; Cesd10, Center for Epidemiologic Studies Depression Scale – 10 items.

high statistical significance ($p < 0.001$). This suggests that the pathogenesis of chronic diseases may involve more complex mechanisms requiring comprehensive consideration of multiple risk factors.

### Relationships of TyG and related indicators with chronic diseases

We next analyzed the dose-response relationships between the TyG index and its related indicators with six chronic diseases (Fig 4). These indices showed positive dose-response relationships with the risks of diabetes, heart disease, dyslipidemia, hypertension, and stroke, though they exhibited different acceleration patterns. Some variables demonstrated exponential or near-linear positive correlation patterns, as seen in the dose-risk relationship between TyG and diabetes (Fig 4B), and between TyG-WHtR and heart disease (Fig 4D). Other variables showed rapid growth within certain ranges, with the growth rate slowing down or even the risk potentially decreasing after reaching certain thresholds, as observed in the dose-risk relationship between TyG-BMI and both heart diseases and stroke (Fig 4A). Interestingly, some indices (TyG, TyG-WC, and WC) showed negative dose-response relationships with digestive system diseases, while others (TyG-BMI, TyG-WHtR, and WHtR) displayed an initial decrease followed by an increase, particularly showing higher risks

**Table 2. Baseline Characteristics of CHARLS Study Participants by Residence Type.**

| Characteristics | Level | Overall (n = 12966) | Urban (n = 4952) | Rural (n = 8014) | P-value |
|---|---|---|---|---|---|
| Age (years) | | 65.00 [57.00, 71.00] | 64.00 [57.00, 71.00] | 65.00 [57.00, 72.00] | 0.060 |
| Gender | Female | 7091 (54.7) | 2783 (56.2) | 4308 (53.8) | 0.007 |
| | Male | 5875 (45.3) | 2169 (43.8) | 3706 (46.2) | |
| Education Level | Below Primary School | 5951 (45.9) | 1672 (33.8) | 4279 (53.4) | <0.001 |
| | Primary School | 2953 (22.8) | 1107 (22.4) | 1846 (23.0) | |
| | Middle School | 2662 (20.5) | 1258 (25.4) | 1404 (17.5) | |
| | High School and Above | 1400 (10.8) | 915 (18.5) | 485 (6.1) | |
| Smoking History | No | 7566 (58.4) | 2974 (60.1) | 4592 (57.3) | 0.002 |
| | Yes | 5400 (41.6) | 1978 (39.9) | 3422 (42.7) | |
| Alcohol Consumption | No | 7086 (54.7) | 2673 (54.0) | 4413 (55.1) | 0.234 |
| | Yes | 5880 (45.3) | 2279 (46.0) | 3601 (44.9) | |
| Body Mass Index (kg/m²) | | 23.75 (3.58) | 24.31 (3.58) | 23.41 (3.54) | <0.001 |
| Hypertension | No | 7633 (58.9) | 2900 (58.6) | 4733 (59.1) | 0.589 |
| | Yes | 5333 (41.1) | 2052 (41.4) | 3281 (40.9) | |
| Diabetes | No | 11329 (87.4) | 4248 (85.8) | 7081 (88.4) | <0.001 |
| | Yes | 1637 (12.6) | 704 (14.2) | 933 (11.6) | |
| Cancer | No | 12591 (97.1) | 4794 (96.8) | 7797 (97.3) | 0.124 |
| | Yes | 375 (2.9) | 158 (3.2) | 217 (2.7) | |
| Pulmonary Disease | No | 10867 (83.8) | 4193 (84.7) | 6674 (83.3) | 0.039 |
| | Yes | 2099 (16.2) | 759 (15.3) | 1340 (16.7) | |
| Dyslipidemia | No | 9652 (74.4) | 3461 (69.9) | 6191 (77.3) | <0.001 |
| | Yes | 3314 (25.6) | 1491 (30.1) | 1823 (22.7) | |
| Cardiovascular Disease | No | 9975 (76.9) | 3663 (74.0) | 6312 (78.8) | <0.001 |
| | Yes | 2991 (23.1) | 1289 (26.0) | 1702 (21.2) | |
| Stroke | No | 11956 (92.2) | 4575 (92.4) | 7381 (92.1) | 0.578 |
| | Yes | 1010 (7.8) | 377 (7.6) | 633 (7.9) | |
| Psychiatric Disorder | No | 12563 (96.9) | 4793 (96.8) | 7770 (97.0) | 0.633 |
| | Yes | 403 (3.1) | 159 (3.2) | 244 (3.0) | |
| Arthritis | No | 7739 (59.7) | 3128 (63.2) | 4611 (57.5) | <0.001 |
| | Yes | 5227 (40.3) | 1824 (36.8) | 3403 (42.5) | |
| Memory-related Disease | No | 12249 (94.5) | 4673 (94.4) | 7576 (94.5) | 0.712 |
| | Yes | 717 (5.5) | 279 (5.6) | 438 (5.5) | |

at lower doses. These findings highlight the complexity and dynamic nature of these indicators in disease risk assessment, suggesting that these factors should be carefully considered when developing intervention strategies.

The study further employed a Cox proportional hazards model to analyze the relationships between different predictor variables and chronic diseases. For each metabolic indicator, participants were divided into terciles (three groups of equal size). Taking TyG as an example, participants were classified based on their TyG values into: Group 1 (TyG < 8.36), Group 2 (8.36 ≤ TyG < 8.87), and Group 3 (TyG ≥ 8.87). The first tercile group (Group 1) served as the reference group (Ref) for comparison with the other two tercile groups. Two adjusted models were simultaneously used to account for potential confounding factors. To address potential residual confounding, we conducted several sensitivity analyses with additional adjustments when data were available, and the results showed robustness. To verify whether the Cox models used in the study satisfied the proportional hazards assumption, tests for the proportional hazard assumption were conducted for all combinations of predictor variables and various chronic diseases (S1 Table). The test results showed differences in the

**Table 3. Provincial Distribution of CHARLS Study Participants and Chronic Disease Prevalence.**

| Province | Participants | Median Age (years) | Male (%) | Hypertension (%) | Diabetes (%) | Cardiovascular (%) | Average Prevalence (%) |
|---|---|---|---|---|---|---|---|
| Shandong | 1,221 | 64.0 | 45.2 | 45.0 | 16.1 | 25.6 | 28.9 |
| Henan | 1,138 | 64.0 | 45.3 | 41.4 | 15.1 | 25.3 | 27.3 |
| Sichuan | 867 | 66.0 | 46.1 | 36.1 | 11.4 | 15.5 | 21.0 |
| Anhui | 784 | 66.0 | 46.9 | 45.4 | 14.7 | 22.8 | 27.6 |
| Yunnan | 779 | 63.0 | 47.4 | 37.2 | 6.8 | 11.9 | 18.7 |
| Jiangxi | 666 | 66.0 | 43.5 | 36.6 | 11.1 | 18.6 | 22.1 |
| Hunan | 633 | 65.0 | 43.8 | 40.1 | 11.8 | 24.6 | 25.5 |
| Inner Mongolia | 590 | 61.0 | 44.6 | 51.5 | 14.2 | 49.7 | 38.5 |
| Hebei | 569 | 65.0 | 45.9 | 53.1 | 16.5 | 36.2 | 35.3 |
| Zhejiang | 560 | 65.0 | 44.8 | 41.4 | 12.1 | 11.4 | 21.7 |
| Jiangsu | 556 | 66.0 | 45.1 | 42.8 | 14.6 | 15.8 | 24.4 |
| Guangdong | 541 | 65.0 | 43.1 | 29.4 | 8.5 | 6.1 | 14.7 |
| Shaanxi | 509 | 63.0 | 49.5 | 36.7 | 10.8 | 27.9 | 25.1 |
| Shanxi | 444 | 63.5 | 47.5 | 43.0 | 13.1 | 20.7 | 25.6 |
| Liaoning | 424 | 63.0 | 44.6 | 43.6 | 11.1 | 29.5 | 28.1 |
| Fujian | 401 | 64.0 | 48.1 | 27.7 | 8.5 | 12.2 | 16.1 |
| Guangxi | 375 | 67.0 | 41.1 | 37.3 | 9.9 | 11.7 | 19.6 |
| Hubei | 349 | 64.0 | 46.7 | 35.2 | 13.8 | 16.9 | 22.0 |
| Gansu | 336 | 63.0 | 44.6 | 45.8 | 14.3 | 28.3 | 29.5 |
| Jilin | 311 | 64.0 | 46.0 | 38.9 | 12.2 | 40.8 | 30.7 |
| Heilongjiang | 263 | 65.0 | 43.0 | 43.0 | 12.5 | 49.0 | 34.9 |
| Chongqing | 169 | 69.0 | 45.6 | 38.5 | 11.8 | 14.8 | 21.7 |
| Guizhou | 129 | 68.0 | 45.0 | 48.1 | 7.0 | 8.5 | 21.2 |
| Tianjin | 102 | 63.0 | 42.2 | 49.0 | 20.6 | 40.2 | 36.6 |
| Qinghai | 96 | 60.0 | 35.4 | 45.8 | 5.2 | 30.2 | 27.1 |
| Xinjiang | 73 | 63.0 | 43.8 | 43.8 | 16.4 | 42.5 | 34.2 |
| Shanghai | 47 | 63.0 | 40.4 | 48.9 | 12.8 | 21.3 | 27.7 |
| Beijing | 34 | 69.5 | 41.2 | 58.8 | 23.5 | 32.4 | 38.2 |

satisfaction of the proportional hazards assumption across different predictor variables and diseases. Specifically, TyG-BMI and BMI satisfied the proportional hazards assumption for most diseases. From the disease perspective, the risk predictions for stroke and digestive system diseases generally met the proportional hazards assumption, while predictions for dyslipidemia mostly violated this assumption. For models that violated the proportional hazards assumption, time-dependent covariate approaches were used for correction to ensure the reliability of the study results.

The study results confirmed the significant positive predictive effects of TyG and its related parameters on chronic disease risks across different tertiles (Table 4). The consistent outcomes from the two models enhance the robustness of our findings. For example, after additional adjustments in Model 2, the hazard ratios of TyG tertile groups for hypertension remained nearly unchanged, indicating that the association between TyG and hypertension is independent of these sociodemographic and lifestyle factors. The same interpretation applies to all other metabolic indicators in the table. Compared to individuals in Tertile 1, those in Tertiles 2 and 3 exhibited a significant positive correlation between these indices and the risk of chronic diseases (adjusted HR > 1, and P < 0.05). Moreover, the HR values for various chronic diseases were higher in the higher TyG Tertile 3 compared to Tertile 2, indicating an increased risk of disease. Notably, in the context of diabetes, although the TyG index itself did not show a statistically significant direct association with diabetes risk in Tertile 2 ($p > 0.05$), but significant

## Chronic Disease Prevalence by Province

Based on CHARLS Data (Hypertension, Diabetes, and Cardiovascular Disease)

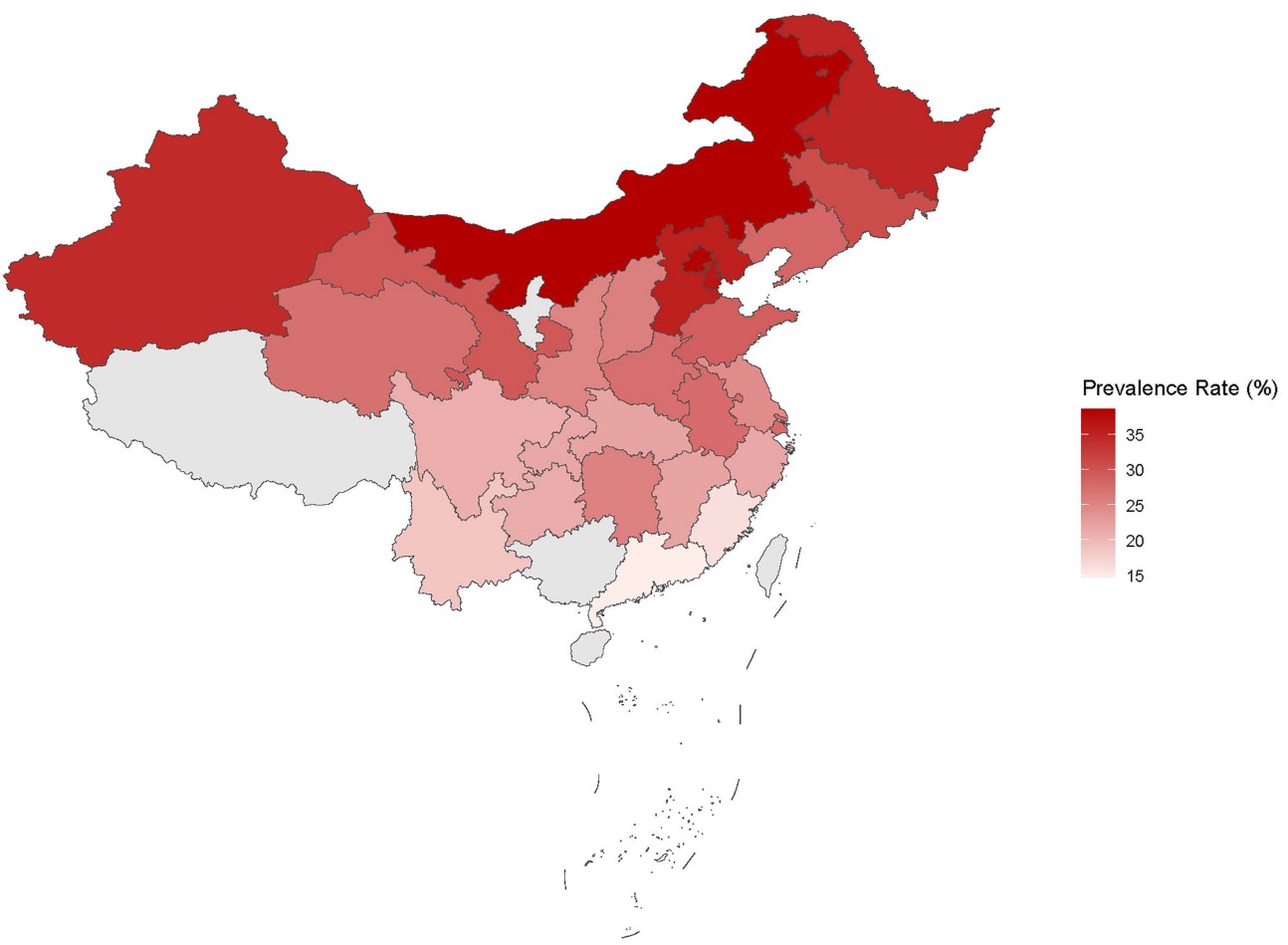

**Fig 1. Chornic disease prevalence by province.** This map was created using data from the USGS National Map Viewer (public domain). Data source: http://viewer.nationalmap.gov/viewer/.

differences were observed in Tertile 3. Meanwhile, other indices such as TyG-BMI, WC, WHtR, TyG-WHtR, and BMI had HR values exceeding 1.5, especially in Tertile 3, these values rose above 4.0. This highlights the crucial role of these indices in diabetes risk assessment. However, there are exceptions. Specifically, in the case of digestive system diseases, most of these parameters showed no significant impact on the incidence of such diseases ($p > 0.05$). Only TyG-BMI, TyG-WC, and BMI were statistically significant ($p < 0.05$) in Tertile 3, and they were associated with a reduced risk of disease. Additionally, the associations between TyG and stroke risk, and between TyG-WC and the risks of heart disease and stroke, did not reach statistical significance ($p > 0.05$), suggesting that other unknown factors may be involved in regulating these diseases.

### Predictive capacity comparison

Next, we evaluated the predictive performance of our model. the predictive performance of TyG and its related parameters for stroke, diabetes, dyslipidemia, heart disease, and hypertension gradually diminished over time (Fig 5A, B, D, E, and F). These

## Distribution of CHARLS Participants by Province
A Geographic Analysis of Study Population

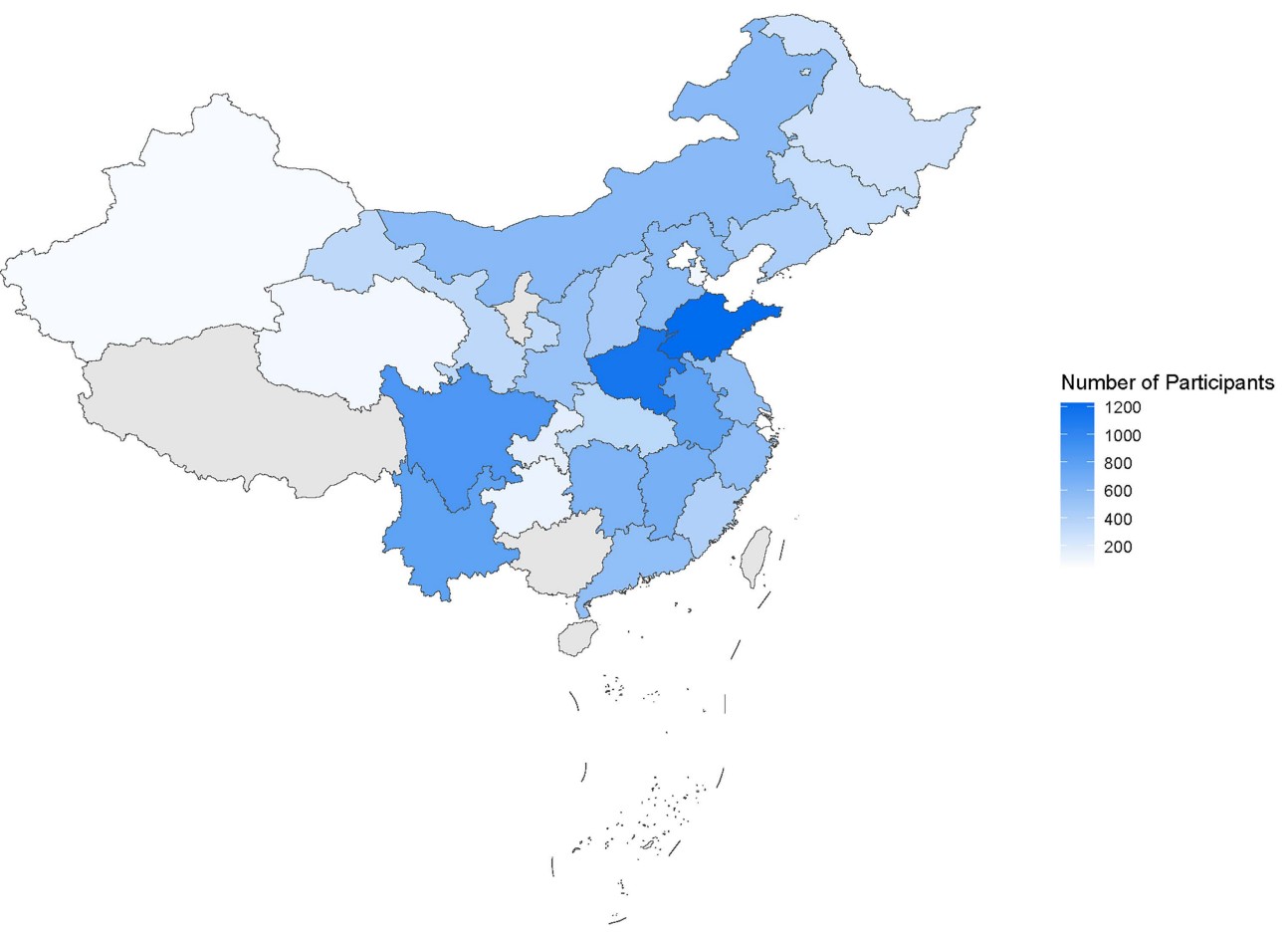

**Fig 2. Distribution of CHARLS participants.** This map was created using data from the USGS National Map Viewer (public domain). Data source: http://viewer.nationalmap.gov/viewer/.

indices exhibited the highest AUC values in the first year. Notably, TyG-WC emerged as a critical predictor, with first-year AUC values reaching 0.678 (95% CI: 0.650–0.760) for stroke, 0.735 (95% CI: 0.720–0.749) for dyslipidemia, 0.664 (95% CI: 0.650–0.678) for heart disease, and 0.707 (95% CI: 0.697–0.716) for hypertension, highlighting its significant role in early disease warning. For diabetes, the overall predictive performance of the TyG index was particularly remarkable, with an initial AUC value as high as 0.771 (95% CI: 0.754–0.788). Despite a slight decline over the eight-year observation period, the AUC values remained stable at 0.731 (95% CI: 0.718–0.744), maintaining a high level of predictive accuracy, this further consolidates its position as the preferred indicator for long-term diabetes monitoring. Similarly, TyG-WC also performed well in long-term diabetes monitoring, with its C-index consistently above 0.7 throughout the 1–8-year period. Regarding hypertension, we observed that compared with other diseases, the predictive performance of TyG and its related parameters showed more pronounced decline over time, suggesting the need for more aggressive intervention measures to curb disease progression. Unlike the diminishing predictive performance seen with other chronic diseases,

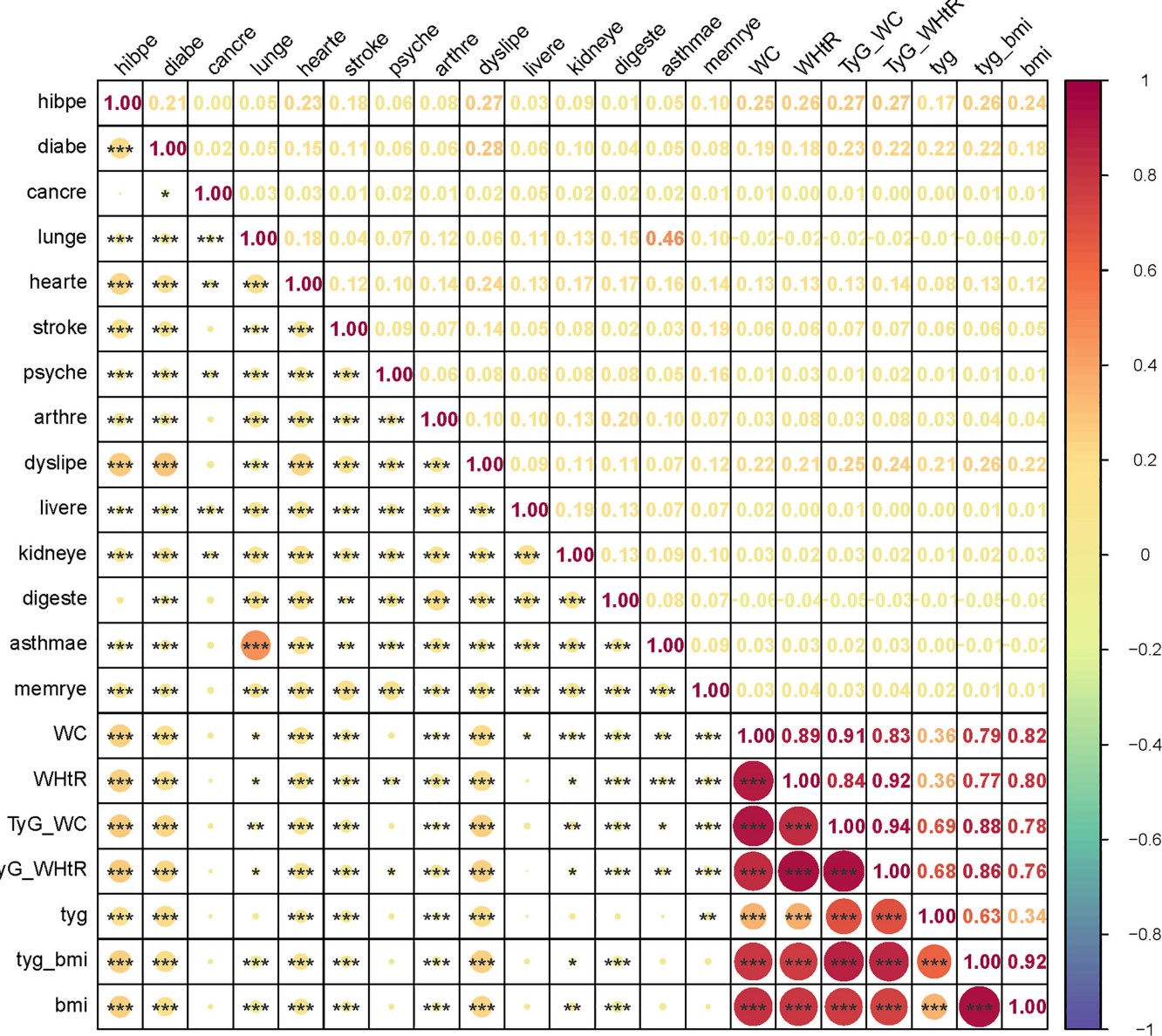

**Fig 3. Pearson's correlation of TyG and related indicators with chronic diseases.** TyG, triglyceride-glucose; BMI, body mass index; WC, waist circumference; WHtR, waist-to-height ratio; diabe, diabetes; hearte, heart disease; dyslipe, dyslipidemia; digeste, digestive disease; arthre, arthritis; dyslipe, dyelipidemia; livere, liver disease; kidneye, kidney disease; digeste, digestive disease; memrye, memory disorder. The color gradient represents the strength of Pearson correlation coefficients (−1 ≤ r ≤ 1), transitioning from purple (negative correlation) to red (positive correlation). Statistical significance is denoted as: *p < 0.05, **p < 0.01, ***p < 0.001.

the predictive power for digestive diseases was relatively weak in the early stages but gradually strengthened over time. However, the overall AUC values remained relatively low (AUC < 0.6) (Fig 5C). Among the predictors, WC stood out as the best indicator, reaching its peak value in the seventh year, though the AUC value was only 0.579 (95% CI: 0.569–0.590).

Among all analyzed metabolic indicators, TyG and TyG-WC exhibited the best discriminative ability for diabetes prediction, with C-indexes of 0.723 (95% CI: 0.716–0.730) and 0.714 (95% CI: 0.707–0.721), respectively, both categorized as "good discrimination" (C ≥ 0.7) (S2 Table). For hypertension, cardiovascular diseases, stroke, and dyslipidemia, most indicators had

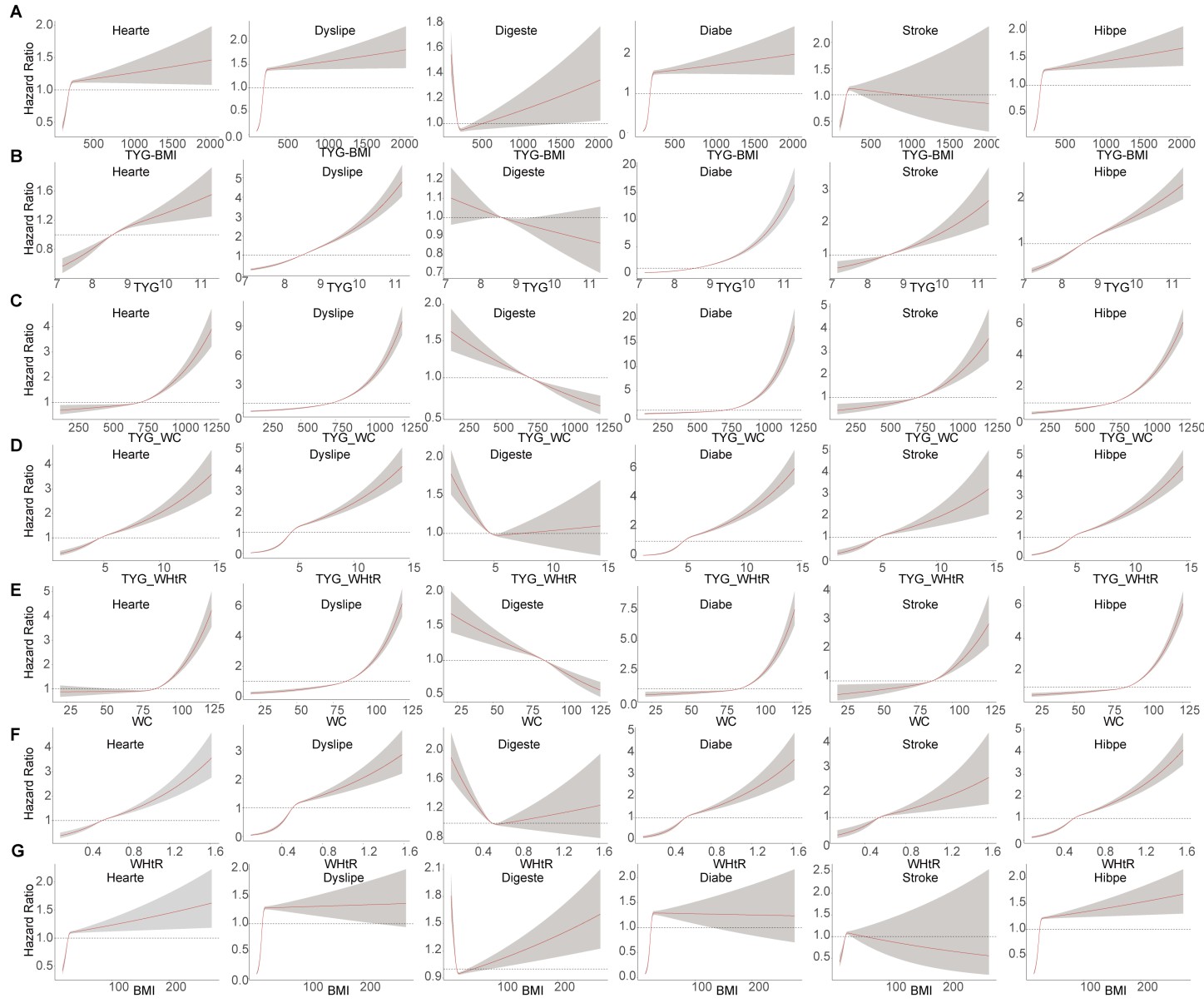

**Fig 4. Dose-responsive relationship of the TyG and related indicators with the risk of chronic diseases.** (A) Dose-responsive relationship of the TyG-BMI and chronic diseases. (B) Dose-responsive relationship of the TyG and chronic diseases. (C) Dose-responsive relationship of the TyG-WC and chronic diseases. (D) Dose-responsive relationship of the TyG-WHtR and chronic diseases. (E) Dose-responsive relationship of the WC and chronic diseases. (F) Dose-responsive relationship of the WHtR and chronic diseases. (G) Dose-responsive relationship of the BMI and chronic diseases. TyG, triglyceride-glucose; BMI, body mass index; WC, waist circumference; WHtR, waist-to-height ratio.

C-indexes between 0.6 and 0.7, falling into the "acceptable discrimination" category. Notably, TyG-WC performed best in predicting hypertension (C = 0.668, 95% CI: 0.661–0.676) and dyslipidemia (C = 0.682, 95% CI: 0.675–0.689). All indicators showed weak performance in predicting digestive system diseases (C < 0.6), which aligns with expectations, as these metabolic metrics may have weaker associations with the pathological mechanisms of digestive disorders. Our models demonstrated varied performance in calibration slopes. The TyG-WHtR had the highest calibration slope (0.373) for predicting

**Table 4. Associations of TyG and related indicators with chronic diseases.**

| Index | Disease | Tertile1 | Tertile2 | | Tertile3 | |
|---|---|---|---|---|---|---|
| | | | HR (95% CI) | P | HR (95% CI) | P |
| TyG_Model1 | hibpe | Ref | 1.150 (1.070-1.237) | <0.001 | 1.324 (1.188-1.476) | <0.001 |
| TyG_Model2 | hibpe | Ref | 1.152 (1.071-1.239) | <0.001 | 1.324 (1.188-1.476) | <0.001 |
| TyG_Model1 | diabe | Ref | 1.142 (0.990-1.318) | 0.068 | 1.399 (1.174-1.667) | <0.001 |
| TyG_Model2 | diabe | Ref | 1.132 (0.981-1.306) | 0.089 | 1.380 (1.157-1.645) | <0.001 |
| TyG_Model1 | hearte | Ref | 1.142 (1.035-1.260) | 0.008 | 1.180 (1.014-1.373) | 0.032 |
| TyG_Model2 | hearte | Ref | 1.130 (1.024-1.247) | 0.015 | 1.157 (0.994-1.347) | 0.059 |
| TyG_Model1 | stroke | Ref | 1.047 (0.888-1.236) | 0.586 | 1.126 (0.882-1.436) | 0.341 |
| TyG_Model2 | stroke | Ref | 1.049 (0.888-1.238) | 0.575 | 1.132 (0.887-1.445) | 0.319 |
| TyG_Model1 | dyslipe | Ref | 1.194 (1.084-1.316) | <0.001 | 1.457 (1.275-1.666) | <0.001 |
| TyG_Model2 | dyslipe | Ref | 1.179 (1.070-1.299) | <0.001 | 1.431 (1.251-1.636) | <0.001 |
| TyG_Model1 | digeste | Ref | 0.985 (0.907-1.070) | 0.719 | 0.988 (0.864-1.130) | 0.862 |
| TyG_Model2 | digeste | Ref | 0.987 (0.908-1.072) | 0.752 | 0.992 (0.867-1.135) | 0.903 |
| TyG_BMI_Model1 | hibpe | Ref | 1.346 (1.263-1.434) | <0.001 | 1.941 (1.829-2.060) | <0.001 |
| TyG_BMI_Model2 | hibpe | Ref | 1.346 (1.264-1.434) | <0.001 | 1.937 (1.824-2.057) | <0.001 |
| TyG_BMI_Model1 | diabe | Ref | 1.832 (1.601-2.096) | <0.001 | 4.848 (4.301-5.463) | <0.001 |
| TyG_BMI_Model2 | diabe | Ref | 1.800 (1.573-2.060) | <0.001 | 4.691 (4.160-5.289) | <0.001 |
| TyG_BMI_Model1 | hearte | Ref | 1.238 (1.139-1.346) | <0.001 | 1.443 (1.332-1.564) | <0.001 |
| TyG_BMI_Model2 | hearte | Ref | 1.213 (1.115-1.319) | <0.001 | 1.381 (1.274-1.498) | <0.001 |
| TyG_BMI_Model1 | stroke | Ref | 1.264 (1.091-1.464) | 0.002 | 1.806 (1.574-2.071) | <0.001 |
| TyG_BMI_Model2 | stroke | Ref | 1.260 (1.088-1.460) | 0.002 | 1.797 (1.565-2.064) | <0.001 |
| TyG_BMI_Model1 | dyslipe | Ref | 1.529 (1.401-1.669) | <0.001 | 2.746 (2.535-2.973) | <0.001 |
| TyG_BMI_Model2 | dyslipe | Ref | 1.491 (1.367-1.628) | <0.001 | 2.620 (2.419-2.839) | <0.001 |
| TyG_BMI_Model1 | digeste | Ref | 0.952 (0.890-1.018) | 0.151 | 0.909 (0.848-0.973) | 0.006 |
| TyG_BMI_Model2 | digeste | Ref | 0.959 (0.896-1.026) | 0.222 | 0.924 (0.863-0.990) | 0.024 |
| WC_Model1 | hibpe | Ref | 1.215 (1.139-1.295) | <0.001 | 1.554 (1.460-1.655) | <0.001 |
| WC_Model2 | hibpe | Ref | 1.220 (1.144-1.300) | <0.001 | 1.564 (1.469-1.666) | <0.001 |
| WC_Model1 | diabe | Ref | 1.693 (1.478-1.939) | <0.001 | 4.090 (3.615-4.627) | <0.001 |
| WC_Model2 | diabe | Ref | 1.675 (1.463-1.918) | <0.001 | 4.018 (3.551-4.547) | <0.001 |
| WC_Model1 | hearte | Ref | 1.158 (1.064-1.260) | <0.001 | 1.248 (1.147-1.357) | <0.001 |
| WC_Model2 | hearte | Ref | 1.142 (1.050-1.243) | <0.001 | 1.214 (1.115-1.320) | <0.001 |
| WC_Model1 | stroke | Ref | 1.185 (1.022-1.374) | 0.024 | 1.561 (1.353-1.801) | <0.001 |
| WC_Model2 | stroke | Ref | 1.185 (1.022-1.375) | 0.024 | 1.562 (1.353-1.804) | <0.001 |
| WC_Model1 | dyslipe | Ref | 1.374 (1.258-1.500) | <0.001 | 2.178 (2.005-2.367) | <0.001 |
| WC_Model2 | dyslipe | Ref | 1.353 (1.239-1.477) | <0.001 | 2.127 (1.957-2.311) | <0.001 |
| WC_Model1 | digeste | Ref | 0.987 (0.922-1.055) | 0.694 | 0.987 (0.922-1.057) | 0.714 |
| WC_Model2 | digeste | Ref | 0.991 (0.926-1.060) | 0.793 | 0.998 (0.932-1.069) | 0.963 |
| TyG_WC_Model1 | hibpe | Ref | 1.086 (1.017-1.160) | 0.014 | 1.139 (1.056-1.227) | <0.001 |
| TyG_WC_Model2 | hibpe | Ref | 1.089 (1.020-1.164) | 0.011 | 1.142 (1.060-1.231) | <0.001 |
| TyG_WC_Model1 | diabe | Ref | 1.446(1.259-1.660) | <0.001 | 2.695 (2.343-3.100) | <0.001 |
| TyG_WC_Model2 | diabe | Ref | 1.437 (1.252-1.650) | <0.001 | 2.679 (2.329-3.082) | <0.001 |
| TyG_WC_Model1 | hearte | Ref | 1.072(0.982-1.169) | 0.119 | 1.010 (0.914-1.117) | 0.840 |
| TyG_WC_Model2 | hearte | Ref | 1.063 (0.975-1.160) | 0.168 | 0.997 (0.902-1.103) | 0.958 |
| TyG_WC_Model1 | stroke | Ref | 1.090 (0.936-1.269) | 0.268 | 1.246 (1.052-1.477) | 0.011 |
| TyG_WC_Model2 | stroke | Ref | 1.090 (0.936-1.269) | 0.267 | 1.247 (1.052-1.478) | 0.011 |
| TyG_WC_Model1 | dyslipe | Ref | 1.20 (1.099-1.316) | <0.001 | 1.497 (1.358-1.651) | <0.001 |

*(Continued)*

**Table 4.** (Continued)

| Index | Disease | Tertile1 | Tertile2 | | Tertile3 | |
|---|---|---|---|---|---|---|
| | | | HR (95% CI) | P | HR (95% CI) | P |
| TyG_WC_Model2 | dyslipe | Ref | 1.192 (1.090-1.306) | <0.001 | 1.489 (1.351-1.643) | <0.001 |
| TyG_WC_Model1 | digeste | Ref | 1.033 (0.964-1.107) | 0.363 | 1.112 (1.028-1.202) | 0.008 |
| TyG_WC_Model2 | digeste | Ref | 1.034 (0.965-1.108) | 0.339 | 1.116 (1.032-1.207) | 0.006 |
| WHtR_Model1 | hibpe | Ref | 1.321 (1.239-1.407) | <0.001 | 1.835 (1.728-1.949) | <0.001 |
| WHtR_Model2 | hibpe | Ref | 1.321 (1.240-1.408) | <0.001 | 1.834 (1.727-1.948) | <0.001 |
| WHtR_Model1 | diabe | Ref | 1.807 (1.579-2.067) | <0.001 | 4.668 (4.140-5.264) | <0.001 |
| WHtR_Model2 | diabe | Ref | 1.776 (1.552-2.033) | <0.001 | 4.526 (4.011-5.106) | <0.001 |
| WHtR_Model1 | hearte | Ref | 1.217 (1.120-1.324) | <0.001 | 1.383 (1.275-1.500) | <0.001 |
| WHtR_Model2 | hearte | Ref | 1.193 (1.097-1.297) | <0.001 | 1.326 (1.221-1.439) | <0.001 |
| WHtR_Model1 | stroke | Ref | 1.241 (1.072-1.438) | 0.004 | 1.713 (1.492-1.966) | <0.001 |
| WHtR_Model2 | stroke | Ref | 1.239(1.069-1.435) | 0.004 | 1.709 (1.487-1.964) | <0.001 |
| WHtR_Model1 | dyslipe | Ref | 1.505 (1.379-1.642) | <0.001 | 2.618 (2.417-2.837) | <0.001 |
| WHtR_Model2 | dyslipe | Ref | 1.468 (1.346-1.602) | <0.001 | 2.504 (2.310-2.714) | <0.001 |
| WHtR_Model1 | digeste | Ref | 0.972 (0.909-1.040) | 0.412 | 0.955 (0.892-1.023) | 0.189 |
| WHtR_Model2 | digeste | Ref | 0.978(0.914-1.047) | 0.522 | 0.970 (0.905-1.038) | 0.376 |
| TyG_WHtR_Model1 | hibpe | Ref | 1.265(1.187-1.348) | <0.001 | 1.642 (1.542-1.747) | <0.001 |
| TyG_WHtR_Model2 | hibpe | Ref | 1.266 (1.188-1.349) | <0.001 | 1.640 (1.540-1.746) | <0.001 |
| TyG_WHtR_Model1 | diabe | Ref | 1.736 (1.517-1.988) | <0.001 | 4.191 (3.705-4.740) | <0.001 |
| TyG_WHtR_Model2 | diabe | Ref | 1.707 (1.491-1.954) | <0.001 | 4.060 (3.588-4.595) | <0.001 |
| TyG_WHtR_Model1 | hearte | Ref | 1.184 (1.088-1.288) | <0.001 | 1.283 (1.177-1.399) | <0.001 |
| TyG_WHtR_Model2 | hearte | Ref | 1.160 (1.066-1.263) | <0.001 | 1.230 (1.127-1.342) | <0.001 |
| TyG_WHtR_Model1 | stroke | Ref | 1.199 (1.035-1.390) | 0.016 | 1.563 (1.352-1.806) | <0.001 |
| TyG_WHtR_Model2 | stroke | Ref | 1.197 (1.033-1.388) | 0.017 | 1.560 (1.348-1.804) | <0.001 |
| TyG_WHtR_Model1 | dyslipe | Ref | 1.448 (1.327-1.581) | <0.001 | 2.364 (2.175-2.568) | <0.001 |
| TyG_WHtR_Model2 | dyslipe | Ref | 1.412 (1.294-1.542) | <0.001 | 2.256 (2.075-2.452) | <0.001 |
| TyG_WHtR_Model1 | digeste | Ref | 1.004 (0.937-1.075) | 0.919 | 1.038 (0.960-1.122) | 0.346 |
| TyG_WHtR_Model2 | digeste | Ref | 1.008 (0.940-1.080) | 0.831 | 1.047 (0.969-1.133) | 0.245 |
| BMI_Model1 | hibpe | Ref | 1.351 (1.268-1.440) | <0.001 | 1.962 (1.848-2.082) | <0.001 |
| BMI_Model2 | hibpe | Ref | 1.351 (1.268-1.440) | <0.001 | 1.958 (1.844-2.079) | <0.001 |
| BMI_Model1 | diabe | Ref | 1.838 (1.607-2.104) | <0.001 | 4.897 (4.346-5.518) | <0.001 |
| BMI_Model2 | diabe | Ref | 1.807 (1.579-2.068) | <0.001 | 4.741 (4.205-5.345) | <0.001 |
| BMI_Model1 | hearte | Ref | 1.241 (1.142-1.350) | <0.001 | 1.452 (1.340-1.574) | <0.001 |
| BMI_Model2 | hearte | Ref | 1.216 (1.118-1.322) | <0.001 | 1.391 (1.283-1.507) | <0.001 |
| BMI_Model1 | stroke | Ref | 1.266 (1.093-1.466) | 0.002 | 1.815 (1.583-2.080) | <0.001 |
| BMI_Model2 | stroke | Ref | 1.262 (1.090-1.462) | 0.002 | 1.807 (1.574-2.073) | <0.001 |
| BMI_Model1 | dyslipe | Ref | 1.535 (1.406-1.675) | <0.001 | 2.774 (2.562-3.003) | <0.001 |
| BMI_Model2 | dyslipe | Ref | 1.497 (1.372-1.634) | <0.001 | 2.648 (2.445-2.869) | <0.001 |
| BMI_Model1 | digeste | Ref | 0.945 (0.888-1.016) | 0.131 | 0.903 (0.845-0.965) | 0.003 |
| BMI_Model2 | digeste | Ref | 0.957 (0.895-1.024) | 0.204 | 0.920 (0.861-0.984) | 0.015 |

TyG, triglyceride-glucose; BMI, body mass index; WC, waist circumference; WHtR, waist-to-height ratio; hibpe, hypertension; diabe, diabetes; hearte, heart disease; dyslipe, dyslipidemia; digeste, digestive disease. Model 1: age and gender were adjusted; Model 2: age, gender, marital status, residential area (urban/rural), education level, smoking status, and alcohol consumption were adjusted.

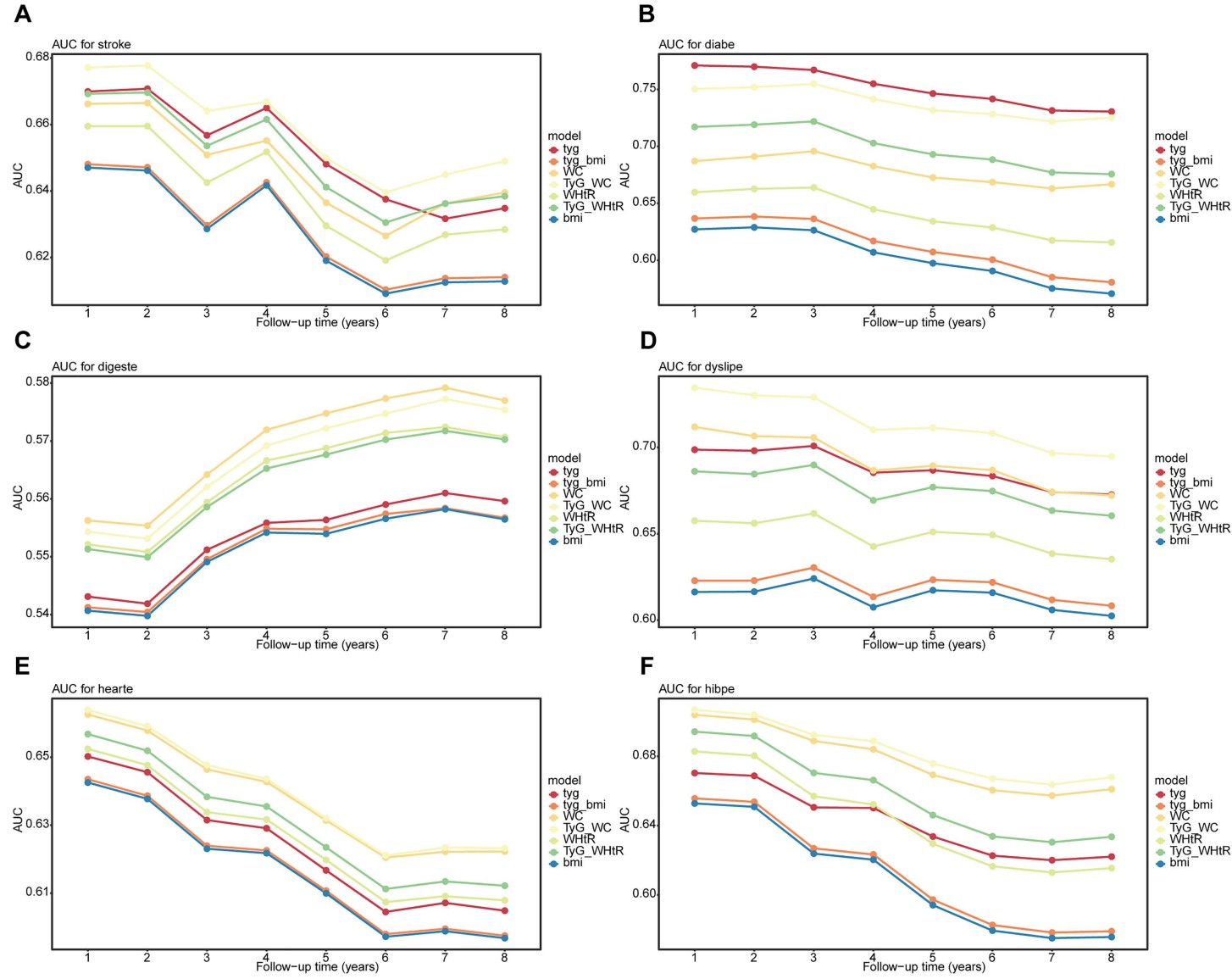

**Fig 5. Time-dependent predictive capacity of TyG and related indicators for chronic diseases.** (A) AUC for stroke. (B) AUC for diabetes. (C) AUC for digestive diseases. (D) AUC for dyslipidemia. (E) AUC for heart disease. (F) hypertension. TyG, triglyceride-glucose; BMI, body mass index; WC, waist circumference; WHtR, waist-to-height ratio.

hypertension, while TyG had the lowest calibration slope (0.065) for predicting stroke. An ideal calibration slope should approach 1.0, and our results indicate a degree of overfitting. Hosmer-Lemeshow test results showed low p-values for most models, suggesting discrepancies between predicted probabilities and observed events. This may be attributed to the use of large cohort data, where even small deviations can lead to statistical significance due to the large sample size.

## Discussion

With the transformation of lifestyles and the rapid development of the economy, especially against the backdrop of the aging population in China, the main threats to human health have gradually shifted from traditional infectious diseases to

chronic diseases. The year-on-year increase in incidence and mortality rates underscores the urgency of disease prevention, early diagnosis, and risk prediction. Previous studies have primarily focused on the associations between the TyG index and its related indices with single categories of diseases such as cardiovascular diseases, chronic kidney disease, or chronic lung disease, or used relatively single predictive indices that cannot comprehensively reflect the relationships with various diseases [27,28,34,35]. In contrast, our study for the first time conducts a large-scale comprehensive analysis of the relationships between multiple chronic diseases and the TyG index and its related indices, aiming to comprehensively characterize the associations between TyG and its related indices with a variety of chronic diseases. Our research findings indicate that the TyG index is significantly correlated with obesity indices and TyG-modified parameters, and there is also a certain correlation with multiple chronic diseases. They exhibit a positive dose-response relationship with the risks of diabetes, heart disease, dyslipidemia, stroke, and hypertension, while showing a negative dose-response relationship or a pattern of first decreasing and then increasing with digestive system diseases. Predictions of model performance show that the TyG index remains the best predictor for diabetes, while TyG-WC exhibits the best predictive performance for hypertension, dyslipidemia, stroke, and heart disease. Over time, the predictive performance of TyG and its related parameters gradually decreases for stroke, diabetes, dyslipidemia, heart disease, and hypertension, while gradually increasing for digestive system diseases—though the predictive value for the latter remains low. These findings are of great significance for the early identification and intervention of chronic disease risks.

The TyG index, obesity indices, and modified TyG indices have been widely used for risk prediction of various diseases. In our study, we combined the TyG index with obesity indices to comprehensively evaluate the associations between these metrics and various chronic diseases. According to our findings, correlation analyses showed relatively weak associations between these indices and multiple chronic diseases. However, in subsequent Cox proportional hazards regression analyses, significant impacts of these metrics on multiple chronic risks were still observed. Most of these parameters were associated with increased chronic risks, which is consistent with most previous studies [27,34–37]. The results showed that most dose-risk response curves exhibited an exponential or approximately linear positive correlation growth pattern, consistent with the findings of Dang [28] and Ren [38]. This pattern reflects the negative impacts of glucose-lipid metabolic disorders and obesity on physical health. However, there are also some high-risk "acceleration areas" or "deceleration areas" associated with the TyG index. This is not uncommon. In the study by Cui et al. [27], a J-shaped relationship was found between the modified TyG index and the incidence of coronary heart disease in the elderly, which they attributed to differences in study design between cohort and cross-sectional frameworks. Lyu et al. [39], discovered a significant inverse "J" shape relationship between TyG-BMI and all-cause mortality in heart failure, and a "U" shape association with heart failure readmission. Qiao et al. [40] suggested that the existence of a J-shaped pattern indicates potential harm to the body from extreme related indicators, and controlling these values around a threshold can help reduce the risk of disease. Differences in research design between cohort and cross-sectional frameworks may explain part of the discrepancies, while participants' age, gender, and region may also influence them. It may be necessary to account for such differences by examining the associations between various diseases and populations in different subgroups. In any case, maintaining a healthy range of TyG index and its related parameters can effectively reduce the risk of chronic diseases. This comprehensive approach to evaluating the TyG index and its related indices provides valuable insights into the risk factors associated with chronic diseases. It emphasizes the importance of considering multiple parameters when assessing an individual's risk for developing these conditions.

Consistent with many studies, TyG demonstrated the best performance in the evaluation of diabetes model performance [41,42], further consolidating its status as the preferred indicator for long-term diabetes monitoring. The good predictive efficacy of these findings may stem from the fact that the TyG index can more directly reflect the core pathophysiological mechanism of diabetes—glucose-lipid metabolic disorders [8]. However, based on a comprehensive assessment of model performance predictions, TyG-WC was the best predictive indicator for these diseases, particularly in stroke, heart disease, hypertension, and dyslipidemia. Although it was not the optimal index for diabetes and digestive

system diseases, its performance remained strong. Therefore, this study suggests that TyG-WC can serve as a comprehensive evaluation parameter for chronic diseases. In addition to its good performance in this study, TyG-WC has also demonstrated strong predictive ability for disease risks in various other contexts, such as non-alcoholic fatty liver disease in Korean adults [43], diabetes in Japanese populations (especially short-term predictions of future diabetes) [44], and metabolic dysfunction-associated fatty liver disease in U.S. populations [45]. The superior performance of TyG-WC may lie in its comprehensive reflection of glucose-lipid metabolism and visceral obesity, which more comprehensively covers the pathological basis of chronic diseases [26].

A rather amazing observation regarding the atypical presentation of digestive diseases was noted in our study. Digestive system diseases demonstrated distinct patterns that differed significantly from other chronic conditions, both in the dose-risk relationship analysis and in model performance evaluation. Digestive system diseases showed a negative dose-response relationship with TyG, TyG-WC, and WC, while demonstrating an initial decrease followed by an increase with TyG-BMI, TyG-WHtR, and WHtR - all of which were associated with higher disease risks at lower levels. Abnormally low TyG index, obesity indices and their modified variants may indicate severely insufficient energy intake or malnutrition, a metabolic state commonly observed in patients with digestive diseases that has been linked to worse disease prognosis and increased mortality risk [46,47]. In the model performance evaluation, these indices showed lower AUC values for digestive diseases compared to other diseases, and interestingly, their AUC values increased over time. These findings suggest we may need to reassess and adjust our monitoring and intervention strategies for digestive system diseases to better align with their unique pathophysiological progression patterns.

The digestive system diseases encompass a broad range, and this risk curve and temporal prediction trend, which differ from other chronic diseases, can be attributed to various factors. These factors include differences in pathophysiological mechanisms, the impact of lifestyle and dietary habits, and even the influence of other underlying diseases more commonly found within this group of diseases. For example, some studies have found that higher TyG indices are positively correlated with the risk of gallstones in digestive system diseases and can serve as predictive indicators for gallstone risk [48,49]. Modified TyG indices are positively associated with fatty liver risk [50]. Larger WC and WHtR are linked to increased incidence of digestive system cancers, with WC showing a stronger correlation with digestive system cancers [51]. However, an analysis by Liu et al. [52] found that BMI and TyG indices do not mediate the association between breakfast frequency and the risk of gastrointestinal cancer. Gao et al.'s research indicated that excessively high or low modified TyG indices may both increase the risk of digestive system diseases [53]. Together with our findings, these results suggest that these indicators are more strongly associated with metabolic disease risks. In the current study, the inclusion of a large number of metabolism-unrelated diseases may dilute the effects of TyG indices and related parameters. In some non-metabolic diseases, metabolic parameters reflecting certain energy reserves may even reduce disease risks. Therefore, future large-sample studies are needed to analyze different subgroups

Long-term physiotherapy care for chronic diseases is a health, psychological, and economic problem that society cannot ignore, and identifying patients' risk factors for chronic diseases is conducive to timely measures for intervention and treatment. However, due to individual differences, variations in research designs, and the presence of various confounding factors, these risk assessments still face many challenges in clinical applications. First, while our data (current through September 2020) captured the initial phase of China's COVID-19 pandemic—enabling analysis of pre/post-pandemic health trends—several limitations should be noted. Pandemic-related home confinement may have reduced physical activity among older adults and disrupted chronic disease management, potentially introducing diagnostic delays and reporting bias [54,55]. Furthermore, generalizability to the "post-pandemic era" is constrained as the data only reflect early-phase metabolic risk changes, not long-term trajectories. Second, the observational design precludes causal inferences between TyG indices (and related parameters) and chronic diseases. Unmeasured confounders—such as occupational exposures and medication use—may influence the results. Systematic data on these factors were unavailable, as occupational details were not collected for retired participants [56,57], and medication records were incomplete. These gaps could affect association

analyses. Third, this study reveals the association between the TyG index and chronic diseases, but it should be noted that there is spatial heterogeneity in sample distribution and disease prevalence. Consistent with previous studies, our results also show that the prevalence of chronic diseases in northern China is higher than that in southern China [58,59]. However, in the sample of this study, the high-prevalence northern provinces of chronic diseases are relatively underrepresented, while the sample proportion of southern provinces with lower prevalence is higher. This discrepancy stems from the PPS sampling method adopted by CHARLS, which leads to a higher proportion of densely populated areas in the sample. Although this sampling method may underestimate the absolute risk in high-prevalence areas to a certain extent, weighted analysis has effectively mitigated potential biases, ensuring the robustness of the research results.

## Conclusion

This study provides significant insights into the relationship between the TyG index and its related parameters and chronic diseases in the elderly population. The findings reveal a positive dose-response relationship between the TyG index and its related parameters with the risk of diabetes, cardiovascular diseases, dyslipidemia, and stroke. Particularly, the TyG-WC demonstrates strong predictive power in the early stages of these diseases, highlighting its utility as an early warning indicator. Interestingly, an inverse relationship was observed between the TyG index and digestive system diseases, suggesting distinct pathophysiological mechanisms at play. The results of our study underscore the complexity of chronic disease development and aid in the identification and risk stratification of diseases using the TyG index and its related parameters, emphasizing the necessity of adopting a multidimensional approach to health assessment.

## Supporting information

**S1 Fig.  Schematic diagram of data preprocessing.**
(TIF)

**S1 Table.  Proportional Hazards Assumption.**
(DOCX)

**S2 Table.  Model Calibration and Discrimination.**
(DOCX)

## Acknowledgments

We express gratitude to the authors of previous studies for depositing their data in CHARLS. Additionally, we extend our appreciation to the researchers who generously shared their R script online, which greatly facilitated our data analysis process.

## Author contributions

**Conceptualization:** Hongli Liu, Xinmei Mao, Xuechen Wang.

**Data curation:** Hongli Liu, Xinmei Mao, Xuechen Wang.

**Formal analysis:** Dan Xu, Ting Wen, Jipeng Li.

**Investigation:** Dan Xu, Ting Wen.

**Methodology:** Dan Xu, Ting Wen.

**Supervision:** Hongli Liu, Xinmei Mao, Xuechen Wang.

**Writing – original draft:** Hongli Liu, Xinmei Mao, Xuechen Wang.

**Writing – review & editing:** Jipeng Li.

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
