## [Decision Letter · Decision Letter 0]

9 May 2025

PONE-D-25-03753Evaluating the Associations and Predictive Performance of Triglyceride-Glucose Index and Related Indicators for Chronic Diseases in a Chinese CohortPLOS ONE

Dear Dr. Li,

Thank you for submitting your manuscript to PLOS ONE. After careful consideration, we feel that it has merit but does not fully meet PLOS ONE’s publication criteria as it currently stands. Therefore, we invite you to submit a revised version of the manuscript that addresses the points raised during the review process.

We look forward to receiving your revised manuscript.

Kind regards,

Debasis Mitra

Academic Editor

PLOS ONE

Journal Requirements:

4. Please ensure that you refer to Figures 1-3 in your text as, if accepted, production will need this reference to link the reader to the figure.

Reviewers' comments:

Reviewer's Responses to Questions

**Comments to the Author**

1. Is the manuscript technically sound, and do the data support the conclusions?

Reviewer #1: Partly

Reviewer #2: Yes

2. Has the statistical analysis been performed appropriately and rigorously? 

Reviewer #1: No

Reviewer #2: No

3. Have the authors made all data underlying the findings in their manuscript fully available?

Reviewer #1: No

Reviewer #2: Yes

4. Is the manuscript presented in an intelligible fashion and written in standard English?

Reviewer #1: Yes

Reviewer #2: Yes

5. Review Comments to the Author

Reviewer #1: The author have worked upon predictive impact of Triglyceride and glucose (TyG) indices for some chronic disease. The author is clear in their objective. However, to provide a more clarity for wider audience, the author is requested to clarify the comments/ suggestions given here somewhere.

The followings are the basic comments need to be clarified:

I. Line no. 22-29. Author has used China Health and 23 Retirement Longitudinal Study (CHARLS) 2011-2020 data in present study. Author need to justify that why old data is taken, it is like timeline of pre-corona condition. Is it going to remain same and pertinent for result in post-covid era? As post-corona time, people facing certain more health issue, so kindly justify it.

II. Line no. 141. Author has mentioned that a total of 12,966 participants were recruited in study from 2011-2020. Author is requested to clear that data was taken just once from each participant or it was taken periodically like after each season, month, year or so on. If it is taken once then timeline of ten years each much longer as during this time many changes like life style, environment, climate change has happened which may influence the results. Justify it.

III. Line no. 141. Author is requested to provide the demographic profile of participant like participant’s, age, sex (male/female/other), educational background, economic status, etc. as this data may influence the status of chronic disease. Here, most important part need to add is (1) participant’s occupation which lead to chronic disease (2) geographical area like state/province as China is very huge in population as well as in geographical area. Author is requested to provide parameters of positive and negative control in present study.

IV. Line no. 144. Author has mentioned that most participants had no history of diabetes or other chronic disease, here author need to specify about the age group to whom this statement is given. As in introduction part (Line no. 47-49), author has mentioned about age group could be 60 years by 2040 which is average age group. But in present study, it is important to know the exact age of participants. Kindly throw some light on it.

V. Line no. 169. Author has explained adjusted model and shown in Table 2. Author need to explain this table in more details. Like raw 1 of shown TyG_Model1 for hibpe disease and then given for Tertile 1 (Ref.), 2, 3. It is very difficult to understand this data. Also, would like to know that how here reference is created. Is this reference different from total population taken? Entire table seems filled with data but difficult understand that how it is calculated, and how to elucidated/conclude the information from this table. Elaborate appropriately.

VI. Line no. 192-204. Author has mentioned that C-value indexed in First year, then C-index given for 1-8 years. Author is requested to provide again clearance about the comparative study has included same individual through 2011-2020 study? If participant were changed then how to make comparative analysis. Please make it clear.

VII. Author has not mentioned about Figure S1, 1, 2, 3 in the text anywhere. Further, author has discussed anything about Figures. The caption give for each figure is not clear. Also, author has mentioned A, B, C and so on sections in figure but not given their details/elaboration in Figure captions. Author is requested work upon it. Author need to explain each figure properly at their justifiable place.

VIII. Line no. 217-220. Author is requested to justify that how they claim that their study is the first, as many reports are available on chronic disease.

IX. Line no. 227-248. Author is requested to elaborate about your results and then give discussion on it.

X. Line no. 249-278. Author has discussed about facts from their results but author need to write that according to Figure/Table or specific data they are observing some facts. Need to re-write scientifically. Further, Line no. 253, author has given a dose-response relationship which is not properly elaborated in result section or material/method section, kindly improve it.

XI. Line no. 280-311. As mentioned in Comment X. author is requested first elaborate about their results and then give supportive reference in discussion.

XII. Author is requested to elaborate the result as well as discussion section thoroughly, at least 3-4 page more to justify your result and discussion part. Author need to explain each figure in details.

XIII. Author is requested to provide your preliminary and secondary data in supplementary documents.

Reviewer #2: The study on association between various TyG related indices to risk of developing chronic diseases is plausible. The study research questions is timely, relevant given to the rise in chronic diseases. However, the manuscript needs some significant and analytical data to make it scientifically rigor. As far as my review i raise the following comments/questions:

1. Have you adjusted key factors like education, income, smoking, diet and physical activity etc? The confounding variables must be adjusted if not to consider the statistical analysis unbiased. Those variables that influence the TyG must be adjusted.

2. There is usage of Cox model in the study. Have you considered the fundamental proportional hazards assumption required for cox model study?? Authors should test this assumption an include in the manucript.

3. Have you evaluated the model calibration in the study to assess discrimination using c index?

4. I would like to raise one doubt, is there any way that elevated TyG can be a indices reflect early stage or preclinical stage of a chronic disease? Have conducted any sensitivity study that has not been mentioned in the manuscript?

5. Make a clear explanation for how the TyG indices is standardized. Provide strong literature review/ discussions to support.

6. PLOS authors have the option to publish the peer review history of their article (what does this mean? ). If published, this will include your full peer review and any attached files.

**Do you want your identity to be public for this peer review?** For information about this choice, including consent withdrawal, please see our Privacy Policy .

Reviewer #1: **Yes: ** Dr. Viralkumar B. Mandaliya

Reviewer #2: **Yes: ** neethu asokan

---

## [Author Response · Author response to Decision Letter 1]

28 May 2025

# Responses to Reviewer Comments

Reviewer 1

We would like to express our gratitude to the reviewers for their comments, and we have made the following revisions accordingly.

Comment 1:

Line no. 22-29. Author has used China Health and 23 Retirement Longitudinal Study (CHARLS) 2011-2020 data in present study. Author need to justify that why old data is taken, it is like timeline of pre-corona condition. Is it going to remain same and pertinent for result in post-covid era? As post-corona time, people facing certain more health issue, so kindly justify it.

Response:

Thank you for your concern regarding our use of CHARLS 2011-2020 data. The China Health and Retirement Longitudinal Study (CHARLS) data is the most recent longitudinal cohort data available to us. The fifth round (2020) data was released in November 2023 and represents the latest dataset from this nationally representative longitudinal study. We appreciate the reviewer’s point about potential changes in health status during the post-pandemic era. Notably, CHARLS’ 2020 survey was conducted in September 2020, which already captured the early stage of the pandemic in China. This provides valuable insights into the health and retirement status of middle-aged and elderly Chinese individuals during this critical transition period.

Additionally, the longitudinal nature of CHARLS (2011–2020) enables us to examine trends over time, which is particularly critical for understanding the trajectory of health issues during the pre-pandemic and early pandemic phases. This temporal perspective offers insights that cannot be obtained from purely post-pandemic analyses. The CHARLS dataset is internationally recognized as a high-quality resource for studying aging in China, and its survey design was approved by the Biomedical Ethics Committee of Peking University (Approval No.: IRB00001052-11015).

We believe that using this comprehensive dataset spanning nearly a decade—including the early pandemic period—provides substantial advantages for understanding the complex health and retirement challenges faced by China’s aging population. Meanwhile, we acknowledge the limitation of lacking more recent post-pandemic data in the discussion section (please refer to lines 567–584).

Comment 2:

Line no. 141. Author has mentioned that a total of 12,966 participants were recruited in study from 2011-2020. Author is requested to clear that data was taken just once from each participant or it was taken periodically like after each season, month, year or so on. If it is taken once then timeline of ten years each much longer as during this time many changes like life style, environment, climate change has happened which may influence the results. Justify it.

Response:

Thank you for raising this important question about our data collection timeline. It is essential to clarify that CHARLS (China Health and Retirement Longitudinal Study) is a longitudinal cohort study designed for periodic follow-ups rather than a single cross-sectional data collection. Our study utilized data from participants recruited at the baseline in 2011, with follow-up data collected across multiple rounds (2013, 2015, 2018, and 2020). The final data used in our analysis were from the fifth round collected in 2020, providing a total follow-up period of approximately 10 years. This longitudinal design is actually a strength of our research, as it allows us to capture changes in health status, lifestyle factors, and environmental influences over time—precisely the concerns highlighted by the reviewer. The 10-year span is not a limitation but enables us to account for the temporal dynamics of these factors and analyze how they impact the health outcomes of middle-aged and elderly Chinese individuals.

CHARLS’ periodic follow-up approach helps document the trajectory of aging in China while considering evolutions in living environments, environmental changes, and medical policy shifts. This methodology is particularly valuable for studying aging-related outcomes, which typically develop gradually over extended periods rather than emerging abruptly. By leveraging this longitudinal dataset with a 10-year follow-up, we can provide more comprehensive insights into the factors influencing the health and retirement of China’s aging population than would be possible with cross-sectional data or shorter follow-up periods.

Comment 3:

Line no. 141. Author is requested to provide the demographic profile of participant like participant’s, age, sex (male/female/other), educational background, economic status, etc. as this data may influence the status of chronic disease. Here, most important part need to add is (1) participant’s occupation which lead to chronic disease (2) geographical area like state/province as China is very huge in population as well as in geographical area. Author is requested to provide parameters of positive and negative control in present study.

Response:

We appreciate the reviewer's valuable suggestions regarding the demographic characteristics and geographic distribution of this study. Your insights are of great significance for enhancing the comprehensiveness and scientific rigor of our research. In accordance with your recommendations, we conducted extensive additional analyses and have supplemented the relevant content in the revised manuscript. Please refer to lines 252–283.

We have added two detailed tables and corresponding analyses:

Table 2: Baseline characteristics stratified by urban/rural residence, including comprehensive information on demographic and clinical features.

Table 3: Distribution of participants across Chinese provinces and chronic disease prevalence rates.

Additionally, we created two geographic distribution maps:

(1) Map of participant numbers across provinces;

(2) Map of geographic distribution of average chronic disease prevalence rates.

These visualizations intuitively illustrate the national distribution characteristics of the CHARLS dataset and the geographic patterns of diseases.

Unfortunately, regarding the occupational information mentioned by the reviewer, the CHARLS questionnaire does not systematically collect detailed occupational data for older adults. As we have already noted this as a limitation in the discussion section (lines 607–614), most participants were retired, making it impossible to include this important variable in our analysis.

Concerning positive and negative controls, this study primarily focuses on the associations between the TyG index and related metrics with multiple chronic diseases, analyzed using the Cox proportional hazards model. We used "digestive system diseases" as a control disease (as shown in Table 4), and the results indicated that most metabolic indicators had no significant associations with digestive system diseases, which partially supports the specificity of our findings.

We are extremely grateful for the reviewer’s valuable suggestions. These additional analyses have significantly enriched the study content and help readers gain a more comprehensive understanding of the regional distribution characteristics and demographic influencing factors of chronic diseases among middle-aged and elderly Chinese populations.

Comment 4:

Line no. 144. Author has mentioned that most participants had no history of diabetes or other chronic disease, here author need to specify about the age group to whom this statement is given. As in introduction part (Line no. 47-49), author has mentioned about age group could be 60 years by 2040 which is average age group. But in present study, it is important to know the exact age of participants. Kindly throw some light on it.

Response:

Thank you for raising this issue. There might be an inaccuracy in our description, so we have revised the presentation in the Results section to clarify the participants' exact age range. Please refer to lines 215–224. The mention in the Introduction about the 2040 age group potentially reaching 60 years old aims to illustrate the increasing aging burden China will face in the future, and the concomitant加重 (escalation) of chronic disease burden, thereby justifying why research on chronic diseases among middle-aged and elderly populations is necessary.

Comment 5:

Line no. 169. Author has explained adjusted model and shown in Table 2. Author need to explain this table in more details. Like raw 1 of shown TyG_Model1 for hibpe disease and then given for Tertile 1 (Ref.), 2, 3. It is very difficult to understand this data. Also, would like to know that how here reference is created. Is this reference different from total population taken? Entire table seems filled with data but difficult understand that how it is calculated, and how to elucidated/conclude the information from this table. Elaborate appropriately.

Response:

Thank you for the reviewer's careful review of Table 4 (original Table 2) and the request for further clarification. We acknowledge that the presentation of our Cox regression analysis results requires additional explanation to enhance understanding.

Table 4 presents hazard ratios (HRs) and 95% confidence intervals (CIs) for the associations between different metabolic indicators (TyG and related indices) and various chronic diseases. The table is organized by indicator type (TyG, TyG-BMI, WC, TyG-WC, WHtR, TyG-WHR, BMI) and disease outcomes (hypertension, diabetes, heart disease, stroke, dyslipidemia, and digestive system diseases).

For each metabolic indicator, participants were divided into terciles (three groups of equal size). Taking TyG as an example, participants were categorized based on their TyG values into:

- Group 1 (TyG < 8.36),

- Group 2 (8.36 ≤ TyG < 8.87),

- Group 3 (TyG ≥ 8.87).

The first tercile (Group 1) served as the reference group (Ref) for comparison with the other two groups.

For instance, in the first row of Table 4 (TyG_Model1 for hypertension), the HR for the second tercile group was 1.150 (95% CI: 1.070–1.237, *P*<0.001), indicating a 15% higher risk of hypertension in the middle TyG tercile compared to the lowest tercile (reference group). Similarly, the third tercile (highest TyG level) showed a 32.4% higher risk (HR=1.324, 95% CI: 1.188–1.476, *P*<0.001).

Two adjusted models were used to account for potential confounders:

Model 1: Adjusted for age and sex only.

Model 2: Adjusted for age, sex, marital status, residence (urban/rural), education level, smoking status, and alcohol consumption.

The consistent results across both models strengthen the robustness of our findings. For example, after additional adjustments in Model 2, the HRs for TyG terciles in hypertension remained nearly unchanged, indicating that the association between TyG and hypertension is independent of these sociodemographic and lifestyle factors. This interpretation applies to all other metabolic indicators in the table.

We hope this clarification aids in understanding the data presented in Table 2. We will add a more detailed explanation of this table in the revised manuscript to improve reader clarity. Please refer to lines 313–165.

Comment 6:

Line no. 192-204. Author has mentioned that C-value indexed in First year, then C-index given for 1-8 years. Author is requested to provide again clearance about the comparative study has included same individual through 2011-2020 study? If participant were changed then how to make comparative analysis. Please make it clear.

Response:

Thank you for the reviewer's important question regarding participant retention in the longitudinal analysis. We wish to clarify that our study design specifically included participants who completed all five rounds of CHARLS surveys from 2011 to 2020. Participants lost to follow-up during this period were excluded from our analysis. This methodological approach was deliberately chosen to ensure the validity of our comparative analyses throughout the study period.

By including only participants with complete follow-up data across all rounds (2011, 2013, 2015, 2018, and 2020), we were able to consistently track the same individuals over the entire 10-year study period. This approach enabled us to calculate accurate C-index values for predicting chronic disease risk from the first to the eighth year of follow-up, as presented in the paper.

Consistently following the same individuals throughout the study period is a key strength of our analysis, as it eliminates potential biases caused by changes in the participant pool. It ensures that comparative analyses of predictive performance (C-index) across time points reflect true changes in prediction accuracy over time, rather than artifacts from participant turnover.

We acknowledge that this restrictive inclusion criterion may have resulted in a smaller sample size compared to analyses that might include participants with incomplete follow-up data. However, we believe this methodological decision strengthens the internal validity of our findings regarding the predictive power of TyG and related indices for chronic disease outcomes over time.

Comment 7:

Author has not mentioned about Figure S1, 1, 2, 3 in the text anywhere. Further, author has discussed anything about Figures. The caption give for each figure is not clear. Also, author has mentioned A, B, C and so on sections in figure but not given their details/elaboration in Figure captions. Author is requested work upon it. Author need to explain each figure properly at their justifiable place.

Response:

Thank you for the reviewer's valuable feedback. We do acknowledge that the description of the figure legends is insufficiently clear. However, due to the fact that some figures contain an excessive number of subplots, it is not appropriate to elaborate on each one individually, and a summary description is more suitable. Otherwise, it would appear rather redundant. However, to enable readers to better understand these figure legends, we have added some detailed descriptions.

Comment 8:

Line no. 217-220. Author is requested to justify that how they claim that their study is the first, as many reports are available on chronic disease.

Response:

Thank you for the reviewer's valuable comments. Our explanation here was indeed unclear, so we have added relevant elaboration to indicate that our study is also the first. Please check lines 435–444.

Comment 9:

Line no. 227-248. Author is requested to elaborate about your results and then give discussion on it.

Response:

Thank you for the reviewer's valuable feedback. Our original intention here was to elaborate on the applications of TyG-related parameters in various studies to demonstrate their importance, which is why we did not discuss them in conjunction with the results. However, during the revision process, we found this to be redundant and somewhat repetitive with the introduction section. Therefore, we deleted this paragraph, retained only part of the content, and re-discussed the results in combination with the following paragraph. Please check lines 461–537.

Comment 10:

Line no. 249-278. Author has discussed about facts from their results but author need to write that according to Figure/Table or specific data they are observing some facts. Need to re-write scientifically. Further, Line no. 253, author has given a dose-response relationship which is not properly elaborated in result section or material/method section, kindly improve it.

Response:

Thank you for the reviewer's valuable suggestions. We have combined the discussion in this paragraph with the previous one and re-discussed the results. Please check lines 461–537. Meanwhile, we have revised the corresponding methodology section and added detailed descriptions in the results section to enhance reader comprehension. Please refer to lines 136–155 and 289–307.

Comment 11:

Line no. 280-311. As mentioned in Comment X. author is requested first elaborate about their results and then give supportive reference in discussion.

Response:

Thank you for the reviewer's valuable comments. We have added result-related elaborations in the discussion and re-discussed them. Please check lines 539–581.

Comment 12:

Author is requested to elaborate the result as we

---

## [Decision Letter · Decision Letter 1]

12 Jun 2025

PONE-D-25-03753R1

Evaluating the Associations and Predictive Performance of Triglyceride-Glucose Index and Related Indicators for Chronic Diseases in a Chinese Cohort

PLOS ONE

Dear Dr. Li,

Thank you for submitting your manuscript to PLOS ONE. After careful consideration, we feel that it has merit but does not fully meet PLOS ONE’s publication criteria as it currently stands. Therefore, we invite you to submit a revised version of the manuscript that addresses the points raised during the review process.

We look forward to receiving your revised manuscript.

Kind regards,

Debasis Mitra

Academic Editor

PLOS ONE

Additional Editor Comments:

Dear Author: reviewers dedicate significant time and effort to provide constructive feedback to enhance the quality and clarity of your manuscript for publication. Taking their comments lightly or failing to address them thoroughly could undermine this process. I strongly recommend carefully addressing all reviewer comments to ensure your manuscript meets the highest standards for publication. Plagiarism report attached.

Reviewers' comments:

Reviewer's Responses to Questions

**Comments to the Author**

1. If the authors have adequately addressed your comments raised in a previous round of review and you feel that this manuscript is now acceptable for publication, you may indicate that here to bypass the “Comments to the Author” section, enter your conflict of interest statement in the “Confidential to Editor” section, and submit your "Accept" recommendation.

Reviewer #1: (No Response)

Reviewer #2: All comments have been addressed

2. Is the manuscript technically sound, and do the data support the conclusions?

Reviewer #1: Partly

Reviewer #2: Yes

3. Has the statistical analysis been performed appropriately and rigorously? 

Reviewer #1: No

Reviewer #2: Yes

4. Have the authors made all data underlying the findings in their manuscript fully available?

Reviewer #1: Yes

Reviewer #2: Yes

5. Is the manuscript presented in an intelligible fashion and written in standard English?

Reviewer #1: Yes

Reviewer #2: Yes

6. Review Comments to the Author

Reviewer #1: The author have revised manuscript and tried to clear various comments. However, there is more scope to revise it and provide a more clarity for wider audience, the author is requested to clarify the comments/ suggestions given here somewhere.

The followings are the basic comments need to be clarified:

1. Regarding Comment 1 in previous communication, author has mentioned that the fifth round (2020) data was released in November 2023 and represents the latest dataset from this nationally representative longitudinal study. Author has mentioned slightly about period covered up to September 2020 in discussion, but it is requested to write in details that why it was took up to this period in data collection part at appropriate place in manuscript.

2. In response to Comment 3, author has created two geographic distribution maps: (1) Map of participant numbers across provinces (Figure 2); and (2) Map of geographic distribution of average chronic disease prevalence rates (Figure 1). From Figure 1, it is apparent that prevalent rate is higher towards northern provinces, but Figure 2 shown that participants surveyed more from southern provinces. Author is requested to clear this dilemma that why no. of participants taken more from southern provinces rather than northerns provinces where rate is alarming.

3. In response to Comment 3, author has mentioned about the occupational information that the CHARLS questionnaire does not systematically collect detailed occupational data for older adults. Author has mentioned that it was noted as a limitation and mentioned in the discussion section (lines 607–614). While searching about line no. 607-614, the below lines observed:

Interestingly, an inverse relationship was observed between the TyG index and digestive system diseases, suggesting distinct pathophysiological mechanisms at play. The results of our study underscore the complexity of chronic disease development and aid in the identification and risk stratification of diseases using the TyG index and its related parameters, emphasizing the necessity of adopting a multidimensional approach to health assessment.

It shows that author has not mentioned proper justified numbering of mentioning in the text. Author is requested to re-write or add this limitation at appropriate place and give exact line number of improvement in manuscript.

4. In response to Comment 4, author has replied that participants' exact age range is clarified and mentioned to refer to lines 215–224. While searching these lines totally different thing is appeared. I suppose assume that author might have changed the font size 14 and because of that line numbering is totally changed. Author again requested to re-write appropriately in manuscript and give exact line number of improvement in manuscript.

5. In response to Comment 5, regarding Table 4, author has cleared and explained it very well, author is requested to write the same in manuscript at appropriate place. As similar to previous comment line no. mentioned regarding this update is not appropriate.

6. In response to Comment 6, it is appreciated that author has taken the data for the individual who is available throughout the study despite reduction in total sample size. Author is requested to write this fact in manuscript at data collection and processing section.

7. In response to Comment 7, author has mentioned about summary description is appropriate for each figure. Suppose, when reader observe Fig 3. Pearson’s correlation of TyG and related indicators with chronic diseases. Regarding this figure, what the * or ** or *** suggesting? What if it highlighted in yellow or red? What is the meaning of scale given right hand side -1 to 1. Further, explanation for Fig 3 given in line no. 280-288 with 82 words is limited. Author is requested that you have worth data then please elaborate it and explain it properly.

8. In continuation to response to Comment 7, regarding Fig 4 is up to author if they wish, as in text Fig 4A, 4B and 4D is mentioned, while in actual figure has 4A to 4G i.e. seven segments. Further, each segment of 4A to 4G has about six line chart (indices). Reader will never be going to understand about this six line chart (indices). It is up to author if they wish elaborate about it or not.

9. While observing the explanation for how the TyG indices is standardized. Author has given justification and supported with seven references. Out of these seven references, three references are not matching/not available.

Like, author has cited,

Fu D, et al. (2019). Triglyceride glucose index in the prediction of adverse cardiovascular events in patients with premature coronary artery disease: a retrospective cohort study. Cardiovasc Diabetol, 18(1), 1-11.

While searching about it, it is found as -

Wu, Z., Liu, L., Wang, W. et al. Triglyceride-glucose index in the prediction of adverse cardiovascular events in patients with premature coronary artery disease: a retrospective cohort study. Cardiovasc Diabetol 21, 142 (2022). https://doi.org/10.1186/s12933-022-01576-8

Another cited reference was

Park B, et al. (2020). Triglyceride glucose index is a risk factor for new-onset type 2 diabetes: A population-based longitudinal study. Diabetes Care, 43(9), 1789-1795.

No, such reference been observed and while searching this journal it was observed that Diabetes Care journal has vol. 43 and issue no. (9) with page no. range 1983-2325.

In another case,

Liu XC, et al. (2022). Triglyceride-glucose index and the risk of hypertension: a meta-analysis of cohort studies. Cardiovasc Diabetol, 21(1), 1-12.

No such reference available.

The citation must be perfect; this kind fictitious citation creates uncertainty on the entire manuscript. Author is instructed to verify all the references in your manuscript, and make ensure that none is fictitious. Further, correct the above citation and write them in manuscript at appropriate place.

Overall, author has taken response to reviewer very casual and replied without giving appropriate numbering. Manuscript could be considered for the next step only after suitable major revision.

Reviewer #2: Seems to have addressed all comments.

If plagiarism also is null then it can be accepted. There were few comments addressed to improve and the authors have tried their part to incorporate the changes 31

7. PLOS authors have the option to publish the peer review history of their article (what does this mean? ). If published, this will include your full peer review and any attached files.

**Do you want your identity to be public for this peer review?** For information about this choice, including consent withdrawal, please see our Privacy Policy .

Reviewer #1: **Yes: ** Dr. Viralkumar B. Mandaliya

Reviewer #2: **Yes: ** Dr Neethu Asokan

---

## [Author Response · Author response to Decision Letter 2]

26 Jun 2025

# Responses to Reviewer Comments

We sincerely appreciate the valuable comments and suggestions from the editors and reviewers, which have significantly contributed to improving our manuscript. In addressing the current round of review comments, we identified several misalignments in the line numbering from our previous response. After careful verification, this discrepancy likely occurred because: while we made secondary formatting adjustments to the manuscript per editorial requests after submitting the revised version, we failed to synchronously update the corresponding line numbers in our response letter. To facilitate more efficient review by the editors and reviewers, we have implemented the following corrective measures:

1.We address all current review comments with all modifications clearly displayed in Track Changes mode;

2.We have annotated our previous revisions using Comments to distinguish them from the current modifications;

3. We have appended our prior response after the current one, with all line numbers now accurately corrected.

We have meticulously detailed all responses and revisions below for your convenience. We deeply apologize for any inconvenience caused by the earlier line numbering errors.

Additionally, please note that Table 4's position is affected when tracked changes are displayed, causing line number discrepancies between the clean manuscript and the version with revision marks. All line numbers referenced in this response letter correspond to the version with revision marks. When the marked manuscript is viewed in "Final Showing Markup" mode, the line numbers between both versions will align consistently.

2025.06.13

Reviewer #1: The author have revised manuscript and tried to clear various comments. However, there is more scope to revise it and provide a more clarity for wider audience, the author is requested to clarify the comments/ suggestions given here somewhere.

Comment 1:

Regarding Comment 1 in previous communication, author has mentioned that the fifth round (2020) data was released in November 2023 and represents the latest dataset from this nationally representative longitudinal study. Author has mentioned slightly about period covered up to September 2020 in discussion, but it is requested to write in details that why it was took up to this period in data collection part at appropriate place in manuscript.

Rosponse

We sincerely appreciate the reviewers' valuable comments. We have supplemented the explanation regarding the time-lagged nature of CHARLS data release in the Data Collection section of our manuscript. As a national longitudinal survey, CHARLS requires an extended period for data collection and processing: The data collected in September 2020 underwent more than three years of cleaning, coding, and quality control before being officially released in November 2023. This timeframe is typical for large-scale social surveys, particularly comprehensive aging studies incorporating complex biometric measurements. We have explicitly stated in the "Data Source and Preprocessing" section that this represents the most current available data from the database. Please check lines 129-131.

Comment 2:

In response to Comment 3, author has created two geographic distribution maps: (1) Map of participant numbers across provinces (Figure 2); and (2) Map of geographic distribution of average chronic disease prevalence rates (Figure 1). From Figure 1, it is apparent that prevalent rate is higher towards northern provinces, but Figure 2 shown that participants surveyed more from southern provinces. Author is requested to clear this dilemma that why no. of participants taken more from southern provinces rather than northerns provinces where rate is alarming.

Rosponse

We sincerely appreciate the reviewer's insightful observation regarding the geographic distribution patterns in our study. Indeed, we have carefully considered the relationship between the sampling distribution and chronic disease prevalence, which reflects both the CHARLS sampling design characteristics and the complexity of chronic disease epidemiology in China.

The CHARLS baseline survey employed a multistage (county/district → village/neighborhood → household) probability-proportional-to-size (PPS) random sampling method with implicit stratification based on region (eastern/central/western), urban-rural status, and per capita GDP. This design resulted in baseline samples distributed across 450 villages/neighborhoods within 150 counties/districts spanning 28 provinces. Importantly, the use of sampling weights has adjusted for differential sampling probabilities across regions, while the original stratification variables (urban/rural, region, GDP) inherently accounted for geographic variations.

The demographic characteristics of CHARLS baseline samples demonstrate strong alignment with the 2010 National Population Census, indicating excellent representativeness of China's middle-aged and elderly population. We have now enhanced the methodological description of sampling procedures in the manuscript and cited the relevant technical report [1] for verification. Please check lines 111-127.

Regarding the north-south prevalence disparity, our findings are consistent with existing literature showing higher chronic disease prevalence in northern China [2,3]. The inverse relationship between sample size and disease prevalence reflects genuine epidemiological patterns rather than sampling bias. We have added explicit discussion of geographic distribution considerations in the Limitations section. Please check lines 630-638.

[1] Zhao, Yaohui, John Strauss, Gonghuan Yang, John Giles, Peifeng (Perry) Hu, et al. (2013). China Health and Retirement Longitudinal Study: 2011-2012 National Baseline User’s Guide, National School of Development, Peking University.

[2] Zhang X, Zhang M, Zhao Z, Huang Z, Deng Q, Li Y, et al. Geographic Variation in Prevalence of Adult Obesity in China: Results From the 2013-2014 National Chronic Disease and Risk Factor Surveillance. Ann Intern Med. 2020 Feb 18;172(4):291-293. doi: 10.7326/M19-0477.

[3] Liu J, Liu M, Chai Z, Li C, Wang Y, Shen M, et al. Projected rapid growth in diabetes disease burden and economic burden in China: a spatio-temporal study from 2020 to 2030. Lancet Reg Health West Pac. 2023 Feb 3;33:100700. doi: 10.1016/j.lanwpc.2023.100700.

Comment 3:

In response to Comment 3, author has mentioned about the occupational information that the CHARLS questionnaire does not systematically collect detailed occupational data for older adults. Author has mentioned that it was noted as a limitation and mentioned in the discussion section (lines 607–614). While searching about line no. 607-614, the below lines observed:

Interestingly, an inverse relationship was observed between the TyG index and digestive system diseases, suggesting distinct pathophysiological mechanisms at play. The results of our study underscore the complexity of chronic disease development and aid in the identification and risk stratification of diseases using the TyG index and its related parameters, emphasizing the necessity of adopting a multidimensional approach to health assessment.

It shows that author has not mentioned proper justified numbering of mentioning in the text. Author is requested to re-write or add this limitation at appropriate place and give exact line number of improvement in manuscript.

Rosponse

We sincerely apologize for any confusion caused. Regarding the limitations concerning occupational information, we have now accurately re-marked the relevant line numbers for your verification. Please check lines 623-630.

Comment 4:

In response to Comment 4, author has replied that participants' exact age range is clarified and mentioned to refer to lines 215–224. While searching these lines totally different thing is appeared. I suppose assume that author might have changed the font size 14 and because of that line numbering is totally changed. Author again requested to re-write appropriately in manuscript and give exact line number of improvement in manuscript.

Rosponse

We sincerely apologize for any confusion caused. We have now carefully verified and corrected all line number references in our response. Please check lines 246-248.

Comment 5:

In response to Comment 5, regarding Table 4, author has cleared and explained it very well, author is requested to write the same in manuscript at appropriate place. As similar to previous comment line no. mentioned regarding this update is not appropriate.

Rosponse

We sincerely appreciate your acknowledgment of our clarification. We deeply regret any confusion caused by the previous line numbering discrepancies and have now carefully re-annotated all relevant line references in the revised manuscript. Please check lines 351-403.

Comment 6:

In response to Comment 6, it is appreciated that author has taken the data for the individual who is available throughout the study despite reduction in total sample size. Author is requested to write this fact in manuscript at data collection and processing section.

Rosponse

We sincerely appreciate the reviewer's valuable comment regarding this issue. In response, we have now supplemented the relevant content in the "Data Source and Preprocessing" section of our manuscript to address this important point. Please check lines 131-134.

Comment 7:

In response to Comment 7, author has mentioned about summary description is appropriate for each figure. Suppose, when reader observe Fig 3. Pearson’s correlation of TyG and related indicators with chronic diseases. Regarding this figure, what the * or ** or *** suggesting? What if it highlighted in yellow or red? What is the meaning of scale given right hand side -1 to 1. Further, explanation for Fig 3 given in line no. 280-288 with 82 words is limited. Author is requested that you have worth data then please elaborate it and explain it properly.

Rosponse

We thank the reviewer for pointing out this issue. Indeed, adding these details will help readers better understand the content. Therefore, we have revised the explanation of Figure 3 by adding more details and included additional annotations below the figure legend. Please check lines 304-324.

Comment 8:

In continuation to response to Comment 7, regarding Fig 4 is up to author if they wish, as in text Fig 4A, 4B and 4D is mentioned, while in actual figure has 4A to 4G i.e. seven segments. Further, each segment of 4A to 4G has about six line chart (indices). Reader will never be going to understand about this six line chart (indices). It is up to author if they wish elaborate about it or not.

Rosponse

We fully understand the reviewer's concerns. However, given that Figure 4 contains multiple subplots with inconsistent dose-response patterns, we have selectively described representative subplots in the Results section to clearly present the key trends to readers. A detailed explanation of every subplot would make the text unnecessarily lengthy and complex, therefore we believe such comprehensive description would not be appropriate.

Comment 9:

While observing the explanation for how the TyG indices is standardized. Author has given justification and supported with seven references. Out of these seven references, three references are not matching/not available.

Rosponse

We sincerely apologize for the mismatched references. We have updated the incorrect references and inserted appropriate citations at relevant positions in the manuscript (listed below). Some references were not suitable for inclusion in the main text and are provided here solely as supplementary information for Reviewer 2's Comment 5 (please verify pages 21-25 of this document). Additionally, we have thoroughly verified all other references in the manuscript to ensure their accuracy.

References:

[1]Zhao, Yaohui, John Strauss, Gonghuan Yang, John Giles, Peifeng (Perry) Hu, et al. (2013). China Health and Retirement Longitudinal Study: 2011-2012 National Baseline User’s Guide, National School of Development, Peking University.

[2]Sánchez-Íñigo L, Navarro-González D, Fernández-Montero A, Pastrana-Delgado J, Martínez JA. The TyG index may predict the development of cardiovascular events. Eur J Clin Invest. 2016 Feb;46(2):189-97. doi: 10.1111/eci.12583.

[3]Cui C, Qi Y, Song J, Shang X, Han T, Han N, et al. Comparison of triglyceride glucose index and modified triglyceride glucose indices in prediction of cardiovascular diseases in middle aged and older Chinese adults. Cardiovasc Diabetol. 2024 May 29;23(1):185. doi: 10.1186/s12933-024-02278-z.

[4]Er LK, Wu S, Chou HH, Hsu LA, Teng MS, Sun YC, et al. Triglyceride Glucose-Body Mass Index Is a Simple and Clinically Useful Surrogate Marker for Insulin Resistance in Nondiabetic Individuals. PLoS One. 2016 Mar 1;11(3):e0149731. doi: 10.1371/journal.pone.0149731.

[5]Guerrero-Romero F, Simental-Mendía LE, González-Ortiz M, Martínez-Abundis E, Ramos-Zavala MG, Hernández-González SO, et al. The product of triglycerides and glucose, a simple measure of insulin sensitivity. Comparison with the euglycemic-hyperinsulinemic clamp. J Clin Endocrinol Metab. 2010 Jul;95(7):3347-51. doi: 10.1210/jc.2010-0288.

[6]Er LK, Wu S, Chou HH, Hsu LA, Teng MS, Sun YC, et al. Triglyceride Glucose-Body Mass Index Is a Simple and Clinically Useful Surrogate Marker for Insulin Resistance in Nondiabetic Individuals. PLoS One. 2016 Mar 1;11(3):e0149731. doi: 10.1371/journal.pone.0149731.

[7]Lopez-Jaramillo P, Gomez-Arbelaez D, Martinez-Bello D, Abat MEM, Alhabib KF, Avezum Á, et al. Association of the triglyceride glucose index as a measure of insulin resistance with mortality and cardiovascular disease in populations from five continents (PURE study): a prospective cohort study. Lancet Healthy Longev. 2023 Jan;4(1):e23-e33. doi: 10.1016/S2666-7568(22)00247-1.

[8]Ren X, Chen M, Lian L, Xia H, Chen W, Ge S, et al. The triglyceride-glucose index is associated with a higher risk of hypertension: evidence from a cross-sectional study of Chinese adults and meta-analysis of epidemiology studies. Front Endocrinol (Lausanne). 2025 Feb 24;16:1516328. doi: 10.3389/fendo.2025.1516328.

[9]Li X, Sun M, Yang Y, Yao N, Yan S, Wang L, et al. Predictive Effect of Triglyceride Glucose-Related Parameters, Obesity Indices, and Lipid Ratios for Diabetes in a Chinese Population: A Prospective Cohort Study. Front Endocrinol (Lausanne). 2022 Mar 30;13:862919. doi: 10.3389/fendo.2022.862919.

2025.05.12

Reviewer 1

We would like to express our gratitude to the reviewers for their comments, and we have made the following revisions accordingly.

Comment 1:

Line no. 22-29. Author has used China Health and 23 Retirement Longitudinal Study (CHARLS) 2011-2020 data in present study. Author need to justify that why old data is taken, it is like timeline of pre-corona condition. Is it going to remain same and pertinent for result in post-covid era? As post-corona time, people facing certain more health issue, so kindly justify it.

Response:

Thank you for your concern regarding our use of CHARLS 2011-2020 data. The China Health and Retirement Longitudinal Study (CHARLS) data is the most recent longitudinal cohort data available to us. The fifth round (2020) data was released in November 2023 and represents the latest dataset from this nationally representative longitudinal study. We appreciate the reviewer’s point about potential changes in health status during the post-pandemic era. Notably, CHARLS’ 2020 survey was conducted in September 2020, which already captured the early stage of the pandemic in China. This provides valuable insights into the health and retirement status of middle-aged and elderly Chinese individuals during this critical transition period.

Additionally, the longitudinal nature of CHARLS (2011–2020) enables us to examine trends over time, which is particularly critical for understanding the trajectory of health issues during the pre-pandemic and early pandemic phases. This temporal perspective offers insights that cannot be obtained from purely post-pandemic analyses. The CHARLS dataset is internationally recognized as a high-quality resource for studying aging in China, and its survey design was app

---

## [Decision Letter · Decision Letter 2]

17 Jul 2025

PONE-D-25-03753R2Evaluating the Associations and Predictive Performance of Triglyceride-Glucose Index and Related Indicators for Chronic Diseases in a Chinese CohortPLOS ONE

Dear Dr. Li,

Thank you for submitting your manuscript to PLOS ONE. After careful consideration, we feel that it has merit but does not fully meet PLOS ONE’s publication criteria as it currently stands. Therefore, we invite you to submit a revised version of the manuscript that addresses the points raised during the review process.

We look forward to receiving your revised manuscript.

Kind regards,

Debasis Mitra

Academic Editor

PLOS ONE

Journal Requirements:

Reviewers' comments:

Reviewer's Responses to Questions

**Comments to the Author**

1. If the authors have adequately addressed your comments raised in a previous round of review and you feel that this manuscript is now acceptable for publication, you may indicate that here to bypass the “Comments to the Author” section, enter your conflict of interest statement in the “Confidential to Editor” section, and submit your "Accept" recommendation.

Reviewer #1: (No Response)

2. Is the manuscript technically sound, and do the data support the conclusions?

Reviewer #1: Partly

3. Has the statistical analysis been performed appropriately and rigorously? 

Reviewer #1: No

4. Have the authors made all data underlying the findings in their manuscript fully available?

Reviewer #1: Yes

5. Is the manuscript presented in an intelligible fashion and written in standard English?

Reviewer #1: Yes

6. Review Comments to the Author

Reviewer #1: Comments/suggestions:

The author have revised manuscript and tried to clear various comments. However, there is more scope to revise it and provide a more clarity for wider audience, the author is requested to clarify the comment/ suggestion given here somewhere.

The following is the basic comment need to be clarified:

1. In response to previous Comment 2 regarding the north-south prevalence disparity, author has mentioned that they had added explicit discussion of geographic distribution considerations in the Limitations section and asked to check lines 630-638. While observing the given lines in both cleared as well as check marked manuscript, the other information/conclusion is observed i.e. it is explicitly not clearly mentioned about north-south prevalence disparity. Author is requested to re-write the particular disparity in manuscript, and for easy understanding please mention the particular line no. either for cleared or check marked manuscript.

Overall, manuscript is suitable after minor revision.

7. PLOS authors have the option to publish the peer review history of their article (what does this mean? ). If published, this will include your full peer review and any attached files.

**Do you want your identity to be public for this peer review?** For information about this choice, including consent withdrawal, please see our Privacy Policy .

Reviewer #1: **Yes: ** Dr. Viralkumar B. Mandaliya

---

## [Author Response · Author response to Decision Letter 3]

27 Jul 2025

# Responses to Reviewer Comments

We sincerely appreciate the reviewers' recognition of our manuscript and their constructive comments for improving its quality. In response to the valuable suggestion, we have made the following revision:

Comment: In response to previous Comment 2 regarding the north-south prevalence disparity, author has mentioned that they had added explicit discussion of geographic distribution considerations in the Limitations section and asked to check lines 630-638. While observing the given lines in both cleared as well as check marked manuscript, the other information/conclusion is observed i.e. it is explicitly not clearly mentioned about north-south prevalence disparity. Author is requested to re-write the particular disparity in manuscript, and for easy understanding please mention the particular line no. either for cleared or check marked manuscript.

Response:

We apologize for the inadequacy of our explanation here. We have rephrased this part to accurately convey the issue of geographical differences and disease prevalence. Please check lines 630-640 in the document check marked manuscript.

---

## [Decision Letter · Decision Letter 3]

6 Aug 2025

Evaluating the Associations and Predictive Performance of Triglyceride-Glucose Index and Related Indicators for Chronic Diseases in a Chinese Cohort

PONE-D-25-03753R3

Dear Dr. Li,

We’re pleased to inform you that your manuscript has been judged scientifically suitable for publication and will be formally accepted for publication once it meets all outstanding technical requirements.

Kind regards,

Debasis Mitra

Academic Editor

PLOS ONE

Additional Editor Comments (optional):

Reviewers' comments:

Reviewer #1: The author have revised manuscript and tried to clear various comments. The revised manuscript is suitable to take forward for the next step.

---

## [Editor Report · Acceptance letter]

PONE-D-25-03753R3

PLOS ONE

Dear Dr. Li,

I'm pleased to inform you that your manuscript has been deemed suitable for publication in PLOS ONE. Congratulations! Your manuscript is now being handed over to our production team.

Kind regards,

on behalf of

Dr. Debasis Mitra

Academic Editor

PLOS ONE